# TLDR: Twin Learning for Dimensionality Reduction

**Yannis Kalantidis**                                    *yannis.kalantidis@naverlabs.com*
*NAVER LABS Europe*

**Carlos Lassance**                                      *carlos.lassance@naverlabs.com*
*NAVER LABS Europe*

**Jon Almazán**                                          *jon.almazan@naverlabs.com*
*NAVER LABS Europe*

**Diane Larlus**                                         *diane.larlus@naverlabs.com*
*NAVER LABS Europe*

**Reviewed on OpenReview:** *https://openreview.net/forum?id=86fhqdBUbx*

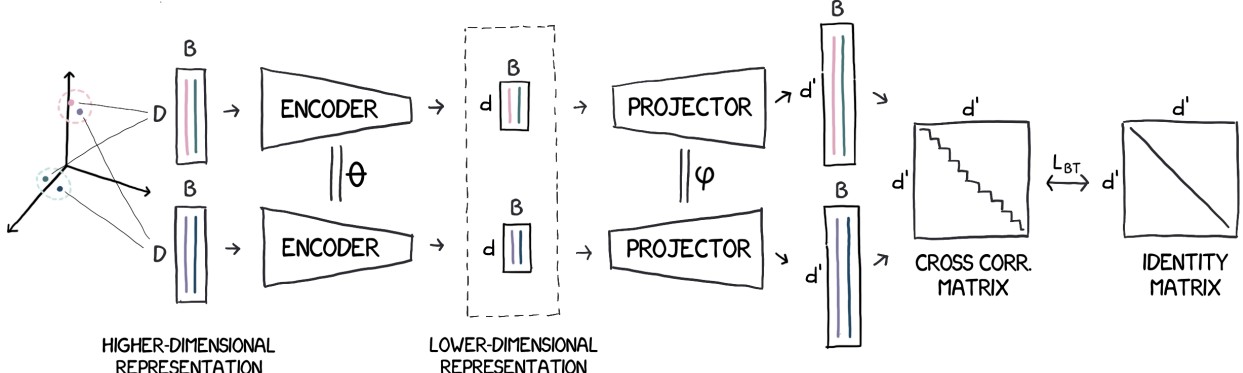

Figure 1: **Overview of the proposed TLDR dimensionality reduction method.** Given feature vectors from a generic input space, we use nearest neighbors to define a set of feature pairs whose proximity we want to preserve. We then learn a dimensionality-reduction function (the *encoder*) by encouraging neighbors in the input space to have similar low dimensional representations using the Barlow Twins loss (Zbontar et al., 2021).

## Abstract

Dimensionality reduction methods are unsupervised approaches which learn low-dimensional spaces where some properties of the initial space, typically the notion of "neighborhood", are preserved. Such methods usually require propagation on large $k$-NN graphs or complicated optimization solvers. On the other hand, self-supervised learning approaches, typically used to learn representations from scratch, rely on simple and more scalable frameworks for learning. In this paper, we propose **TLDR**, a dimensionality reduction method for generic input spaces that is porting the recent self-supervised learning framework of Zbontar et al. (2021) to the specific task of dimensionality reduction, over arbitrary representations. We propose to use nearest neighbors to build pairs from a training set and a redundancy reduction loss to learn an encoder that produces representations invariant across such pairs. TLDR is a method that is simple, easy to train, and of broad applicability; it consists of an offline nearest neighbor computation step that can be highly approximated, and a straightforward learning process. Aiming for scalability, we focus on improving *linear* dimensionality reduction, and show consistent gains on image and document retrieval tasks, *e.g.* gaining $+4\%$ mAP over PCA on $\mathcal{R}$Oxford for GeM-AP, improving the performance of DINO on ImageNet or retaining it with a $10\times$ compression.

Code available at: https://github.com/naver/tldr

## 1 Introduction

Self-supervised representation learning (SSL) has been shown to produce representations that are highly transferable to a wide number of downstream tasks via encoding invariance to image distortions (Chen et al., 2020a; He et al., 2020; Caron et al., 2020). Methods like BYOL (Grill et al., 2020), DINO (Caron et al., 2021) or Barlow Twins (Zbontar et al., 2021) start from *structured inputs* (i.e. a 2D grid of pixel intensities) and learn *a large encoder* with hundreds of millions of parameters that transforms this input into a representation vector useful for downstream tasks. Central to their success is the scalable and easy-to-optimize learning framework that such methods adopt, often based on pairwise loss functions with or without constrasting pairs.

Dimensionality reduction refers to a set of (generally unsupervised) approaches which aim at learning low-dimensional spaces where properties of an initial higher-dimensional input space, *e.g.* proximity or "neighborhood", are preserved. Unlike SSL, dimensionality reduction starts from *any* trustworthy, potentially blackbox, representation and learns a *simple* (often linear) encoder without any further assumptions on the nature of the input space beyond the fact that structure should be preserved. Dimensionality reduction is *a crucial component* in many areas as diverse as biology or AI, and used for tasks ranging from visualization and compression, to indexing and retrieval. This is because, more often than not, learning a small encoder on top of an existing representation is easier than training a lower dimensionality representation end-to-end; in many cases, training from scratch or fine-tuning a representation is too costly or simply not possible as features might be produced by a blackbox module or sensor. Considering how manifold learning methods lack scalability and usually require propagation on large $k$-NN graphs or complicated optimization solvers, it is only natural to wonder: *Can we borrow from the highly successful learning frameworks of self-supervised representation learning to design dimensionality reduction approaches?*

In this paper, we propose *Twin Learning for Dimensionality Reduction* or **TLDR**, a generic dimensionality-reduction technique where the only prior is that data lies on a reliable manifold we want to preserve. It is based on the intuition that comparing a data point and its nearest neighbors is a good "distortion" to learn from, and hence a good way of approximating the local manifold geometry. Similar to other manifold learning methods (Roweis & Saul, 2000; Van der Maaten & Hinton, 2008; Belkin & Niyogi, 2003; Donoho & Grimes, 2003; Hadsell et al., 2006) we use Euclidean nearest neighbors as a way of defining distortions of the input that the dimensionality reduction function should be invariant to. However, unlike other manifold learning methods, TLDR does not require eigendecompositions, negatives to contrast, or cumbersome optimization solvers; it simply consists of an offline nearest neighbor computation step that can be highly approximated without loss in performance and a straightforward stochastic gradient descent learning process. This leads to a highly scalable method that can learn linear and non-linear encoders for dimensionality reduction while trivially handling out-of-sample generalization. We show an overview of the proposed method in Figure 1.

TLDR is meant as a post-hoc dimensionality reduction method, *i.e. on top of* learned representations. We show that it provides an easy way of not only compressing representations, but also improving the performance of large state-of-the-art models relying on those reduced representations, *without the need to fine-tune* large encoders. Training end-to-end or fine-tuning requires large resources, specially when starting from large pre-trained models composed of millions or billions of parameters, and application to certain domains or tasks requires non-trivial know-how. Those issues fade when large models are treated as feature extractors, and we only learn a linear or MLP encoder to compress those features vectors in a representation that is more suited to the desired task or dataset. This highlights the natural link between representation learning and dimensionality reduction: we could use the output of models like Barlow Twins, BYOL, DINO, BERT or any representation learning method as the input for our TLDR encoder. Yet, those two steps are not interchangeable and we focus on the latter.

Aiming at large-scale search applications, we focus on improving *linear* dimensionality reduction with a compact encoder, an integral part of the first-stage of most retrieval systems, and an area where PCA (Pearson, 1901) is still the default method used in practice (Tolias et al., 2020; Weinzaepfel et al., 2022). We present a large set of ablations and experimental results on common benchmarks for image retrieval, as well as on the *natural language processing* task of argument retrieval. We show that one can achieve significant gains without altering the encoding and search complexity: for example we can improve landmark image retrieval with GeM-AP (Revaud et al., 2019) on ROxford5K (Radenović et al., 2018a) by almost *4 mAP*

*points* for 128 dimensions, a commonly used dimensionality, by simply replacing PCA with a linear TLDR encoder. Similarly, we are able to improve the state-of-the-art retrieval performance of DINO (Caron et al., 2021) representations on ImageNet (Russakovsky et al., 2015), even when compressing the vectors tenfold.

We also perform extensive evaluations using DINO models with a ViT backbone and report state-of-the-art retrieval results for fully self-supervised learning methods on multiple datasets like $\mathcal{R}$Oxford, $\mathcal{R}$Paris, and ImageNet. We further show that, given access to unlabeled data from the downstream domain, simply learning a linear TLDR encoder on top of generic DINO representations leads to large performance gains on the downstream task, without the need to fine-tune the backbone. Finally, we show that TLDR is robust to both approximate neighbor pair generation, as well as subsequent vector quantization.

**Contributions.** We introduce TLDR, a dimensionality reduction method that achieves neighborhood embedding learning with the simplicity and effectiveness of recent self-supervised visual representation learning losses. Aiming for scalability, we focus on large-scale image and document retrieval where dimensionality reduction is still an integral component. We show that replacing PCA (Pearson, 1901) with a *linear* TLDR encoder can greatly improve the performance of state-of-the-art methods without any additional computational complexity. We thoroughly ablate parameters and show that our design choices allow TLDR to be robust to a large range of hyper-parameters and is applicable to a diverse set of tasks and input spaces.

## 2 Twin Learning for Dimensionality Reduction

Starting from a set of unlabeled and high-dimensional features, our goal is to learn a lower-dimensional space which preserves the local geometry of the larger input space. Assuming that we have no prior knowledge other than the reliability of the local geometry of the input space, we use nearest neighbors to define a set of feature pairs whose proximity we want to preserve. We then learn the parameters of a dimensionality-reduction function (the *encoder*) using a loss that encourages neighbors in the input space to have similar representations, while also minimizing the redundancy between the components of these vectors. Similar to other works (Chen et al., 2020b;a; Zbontar et al., 2021) we append a *projector* to the encoder that produces a representation in a very high dimensional space, where the Barlow Twins (Zbontar et al., 2021) loss is computed. At the end of the learning process, the projector is discarded. All aforementioned components are detailed next. We call our method **Twin Learning for Dimensionality Reduction** or **TLDR**, in homage to the Barlow Twins loss. An overview is provided in Figure 1 and in Algorithm 1.

**Preserving local neighborhoods.** Recent self-supervised learning methods define positive pairs via hand-crafted distortions that exploit prior information from the input space. In absence of any such prior knowledge, defining local distortions can only be achieved via assumptions on the input manifold. Assuming a locally linear manifold, for example, would allow using the Euclidean distance as a local measure of on-manifold distortion and using nearest neighbors over the training set would be a good approximation for local neighborhoods. Therefore, we construct pairs of neighboring training vectors, and learn invariance to the distortion from one such vector to another. Practically, we define the local neighborhood of each training sample as its $k$ nearest neighbors. Although defining local neighborhood in such a uniform way over the whole manifold might seem naive, we experimentally show that not only it is sufficient, but also that our algorithm is robust across a wide range of values for $k$ (see Section 3.2).

Using nearest neighbors is of course not the only way of defining neighborhoods. In fact, we show alternative results with a simplified variant of TLDR (denoted as TLDR$_\mathcal{G}$) where we construct pairs by simply adding Gaussian noise to an input vector. This is a baseline resembling denoising autoencoders, although in our case we are using a) an asymmetric encoder-decoder architecture and b) the Barlow twins loss instead of a reconstruction loss.

**Notation.** Our goal is to learn an encoder $f_\theta : \mathbb{R}^D \to \mathbb{R}^d$ that takes as input a vector $x \in \mathbb{R}^D$ and outputs a corresponding reduced vector $z = f_\theta(x) \in \mathbb{R}^d$, with $d << D$. Without loss of generality, we define the encoder to be a neural network with learnable parameters $\theta$. Let $\mathcal{X}$ be a (training) set of datapoints in $\mathbb{R}^D$, the $D$-dimensional input space. Let $x \in \mathbb{R}^D$ be a vector from $\mathcal{X}$. $\mathcal{N}_k(x)$ is composed of the $k$ nearest neighbors of $x$. For a vector $y \in \mathcal{X}$ from the training set: $y \in \mathcal{N}_k(x) \Leftrightarrow y \in \arg_k \min_{y \in \mathcal{X}} d(x, y)$, where $d(\cdot, \cdot)$ denotes the Euclidean distance. Although the definition above can be trivially extended to non-Euclidean

---

**Algorithm 1:** Twin Learning for Dimensionality Reduction (TLDR)

---

**Input:** Training set of high-dimensional vectors $\mathcal{X}$, hyper-parameter $k$
**Output:** Corresponding set of lower-dimensional vectors $\mathcal{Z}$,
   dimensionality-reduction function $f_\theta : \mathbb{R}^D \to \mathbb{R}^d$
1: Calculate the $k$ (approximate) nearest neighbors, *i.e.* $\mathcal{N}_k(x)$, for every $x \in \mathcal{X}$.
2: Create positive pairs $(x, y)$ by sampling $y$ from the set $\mathcal{N}_k(x)$.
3: Learn the parameters $\theta$ of $f_\theta$ and $\phi$ of $g_\phi$ by optimizing the Barlow Twins' loss (Eq. (1)).

---

distances and adaptive neighborhoods (*e.g.* defined by a radius), without loss of generality we present our method and results with pairs from $k$ Euclidean neighbors. We define *neighbor pairs* as pairs $(x, y) \in \mathcal{X} \times \mathcal{X}$ where $y \in \mathcal{N}_k(x)$.

**Learning à la Barlow Twins.** Although contrastive losses were proven highly successful for visual representation learning, explicitly minimizing the redundancy of the output dimension is highly desirable for dimensionality reduction: having a highly informative output space is more important than a highly discriminative one. We therefore choose to learn the parameters of our encoder by minimizing the Barlow Twins loss function (Zbontar et al., 2021), C: which suits C: the problem perfectly. Similar to (Zbontar et al., 2021), we append a projector $g_\phi$ to the encoder $f_\theta$, allowing to calculate the loss in a (third) representation space which is not the one that will be used for subsequent tasks. That extended space can possibly be much larger. We detail the encoder and the projector later. Let $\hat{z} = g_\phi(f_\theta(x))$ be the output vector of the projector, $\hat{z} \in \mathbb{R}^{d'}$. Given a pair of neighbors $(x^A, x^B)$ and the corresponding vectors $\hat{z}^A, \hat{z}^B$ after the projector, the loss function $\mathcal{L}_{BT}$ is given by:

$$\mathcal{L}_{BT} = \sum_i (1 - \mathcal{C}_{ii})^2 + \lambda \sum_i \sum_{i \neq j} \mathcal{C}_{ij}^2, \quad \text{where } \mathcal{C}_{ij} = \frac{\sum_b \hat{z}_{b,i}^A \hat{z}_{b,j}^B}{\sqrt{\sum_b (\hat{z}_{b,i}^A)^2} \sqrt{\sum_b (\hat{z}_{b,j}^B)^2}}, \tag{1}$$

where $b$ indexes the positive pairs in a batch, $i$ and $j$ are two dimensions from $\mathbb{R}^{d'}$ (*i.e.* $0 \leq i, j \leq d'$) and $\lambda$ is a hyper-parameter. $\mathcal{C}$ is the $d' \times d'$ cross-correlation matrix computed and averaged over all positive pairs $(\hat{z}^A, \hat{z}^B)$ from the current batch. The loss is composed of two terms. The first term encourages the diagonal elements to be equal to 1. This makes the learned representations invariant to applied distortions, *i.e.* the datapoints moving along the input manifold in the neighborhood of a training vector are encouraged to share similar representations in the output space. The second term is pushing off-diagonal elements towards 0, reducing the redundancy between output dimensions, a highly desirable property for dimensionality reduction.

The redundancy reduction term can be viewed as a soft-whitening constraint on the representations and, as shown in Zbontar et al. (2021), it works better than performing "hard" whitening on the representations (Ermolov et al., 2021). Finally, it is worth noting that understanding the dynamics of learning without contrasting pairs is far from trivial and beyond the scope of this paper; we refer the reader to the recent work by Tian et al. (2021) that studies this learning paradigm in depth and discusses why trivial solutions are avoided when learning without negatives as in Eq. (1).

**The encoder $f_\theta$.** We consider a number of different architectures for the encoder:

- *linear*: The most straight-forward choice for encoder $f_\theta$ is a linear function parametrized by a $D \times d$ weight matrix $W$ and bias term $b$, *i.e.* $f_\theta(x) = Wx + b$. Beyond computational benefits, and given that we are mostly interested in medium-sized output spaces where $d \in \{8, \ldots, 512\}$, we argue that, given a meaningful enough input space, a linear encoder could suffice in preserving neighborhoods of the input.

- *factorized linear*: Exploiting the fact that batch normalization (BN) (Ioffe & Szegedy, 2015) is linear during inference,[1] we formulate $f_\theta$ as a multi-layer linear model, where $f_\theta$ is a sequence of $l$ layers, each composed of a linear layer followed by a BN layer. This model introduces non-linear dynamics which can

---

[1]Although batch normalization is non-linear during training because of the reliance on the current batch statistics, during inference and using the means and variances accumulated over training, it reduces to a linear scaling applied to the features, that can be embedded in the weights of an adjacent linear layer.

potentially help during training but the sequence of layers can still be replaced with a single linear layer after training for efficiently encoding new features.

- *MLP*: $f_\theta$ can be a multi-layer perceptron with batch normalization (BN) (Ioffe & Szegedy, 2015) and rectified linear units (reLUs) as non-linearities, *i.e.* $f_\theta$ would be a sequence of $l$ linear-BN-reLU triplets, each with $H^i$ hidden units $(i = 1, .., l)$, followed by a linear projection.

Our main goal is to develop a scalable alternative to PCA for dimensionality reduction, so we are mostly interested in linear and factorized linear encoders. It is worth already mentioning that, as we will show in our experimental validation, gains from introducing an MLP in the encoder are minimal and would not justify the added computational cost in practice.

**The projector $g_\phi$.** As also recently noted in Tian et al. (2021), a crucial part of learning with non-contrastive pairs is the projector. This module is present in a number of contrastive self-supervised learning methods (Chen et al., 2020a; Grill et al., 2020; Zbontar et al., 2021; Tian et al., 2021). It is usually implemented as an MLP inserted between the transferable representations and the loss function. Unlike other methods, however, where the projector takes the representations to an even lower dimensional space for the contrastive loss to operate on (*i.e.* for SimCLR (Chen et al., 2020a) and BYOL (Grill et al., 2020), $d' \ll d$), for the Barlow Twins objective, operating in large output dimensions is crucial.

In Section 3, we study the impact of the dimension $d'$ and experiment with a wide range of values. We empirically verify the findings of Zbontar et al. (2021) that calculating the de-correlation loss in higher dimensions ($d' \gg d$) is highly beneficial. In this case, and as shown in Figure 1, the transferable representation is now the *bottleneck* layer of this non-symmetrical hour-glass model. Although Eq. (1) is applied after the projector and only indirectly decorrelates the output representation components, having more dimensions to decorrelate leads to a representation that is more informative: the *bottleneck* effect created by the projector's output being in a much larger dimensionality implicitly enables the network to learn an encoder that also has more decorrelated outputs.

## 3 Experimental validation

In this section, we present a set of experiments validating the proposed TLDR on both visual and textual representations. In Section 3.1 we first present a summary of the tasks we evaluate on. Then in Sections 3.2 and 3.3 we present results for visual and textual applications, respectively. Finally, in Section 3.4 we study the different hyper-parameters of our method, compare to several manifold learning methods, and present results when approximating the nearest neighbor pairs, as well as results after subsequent vector quantization.

### 3.1 Summary of tasks and evaluation protocol

A summary of all the tasks, datasets and representations that we consider in this section is presented in Table 1. We explore tasks like landmark image retrieval on $\mathcal{R}$Oxford and $\mathcal{R}$Paris, object class retrieval on ImageNet, and argument retrieval on ArguAna. For all experiments, we start from reliable feature vectors; it is a pre-requirement for the dimensionality reduction task. We assume that any structured data (images or documents) is first encoded with a suitable representation, and, without loss of generality, we assume the Euclidean distance to be meaningful, at least locally, in this input representation space.

In Section 3.2 we evaluate TLDR on popular image retrieval tasks like *landmark image retrieval* on datasets like $\mathcal{R}$Oxford, $\mathcal{R}$Paris (Radenović et al., 2018a), or $k$-NN retrieval on ImageNet (Russakovsky et al., 2015). We start from both specialized, retrieval-oriented representations like GeM-AP (Revaud et al., 2019), as well as more generic representations learned via self-supervised learning like DINO (Caron et al., 2021). For all experiments, we use the DINO and AP-GeM models as feature extractors to encode images for visual tasks; we use global image features from publicly avalable models for GeM-AP[2] or DINO[3]. For the textual domain, we focus on the task of *argument retrieval* (Section 3.3). We use 768-dimensional features from an

---

[2]https://github.com/naver/deep-image-retrieval
[3]https://github.com/facebookresearch/dino

Table 1: **Tasks, models and summary for *linear* dimensionality reduction.** [†] denotes the use of the dataset without labels. The GeM-AP and DINO models are from Revaud et al. (2019) and Caron et al. (2021), respectively. Different values of $d$ for TLDR and PCA are denoted in parenthesis next to the TLDR performance when $d \neq 128$. [‡] input space $D = 384$ for ViT-S/16; [§] input space $D = 768$ for BERT.

| Task | Retrieval dataset (Metric) | Representation | | Dim. red. | Result Summary | | |
|------|------|------|------|------|------|------|------|
| | | Model | Dataset | dataset | TLDR $(d = 128)$ | vs PCA$_w$ $(d = 128)$ | vs Orig. $(D = 2048)$ |
| Landmark Retrieval | $\mathcal{R}$Oxford (*mAP*) | GeM-AP (ResNet-50) | Landmarks | GLD-v2 [†] | 0.49 | ↑ **4.1%** | ↑ **0.7%** |
| | | DINO (ResNet-50) | ImageNet [†] | ImageNet [†] | 0.22 | ↑ **2.9%** | ↓ **0.8%** |
| | | | | GLD-v2 [†] | 0.29 | ↑ **3.7%** | ↑ **6.8%** |
| | | DINO (ViT-S/16) | ImageNet [†] | ImageNet [†] | 0.25 | ↑ **2.6%** | ↓ **0.2%**[‡] |
| | | | | GLD-v2 [†] | 0.31 | ↑ **2.7%** | ↑ **5.4%**[‡] |
| | | | GLD-v2 [†] | GLD-v2 [†] | 0.43 (256) | ↑ **1.9%** | ↑ **4.6%**[‡] |
| | $\mathcal{R}$Paris (*mAP*) | GeM-AP (ResNet-50) | Landmarks | GLD-v2 [†] | 0.67 | ↑ **1.2%** | ↑ **0.6%** |
| ImageNet Retrieval | ImageNet (*k*-NN *Top-1 Acc.*) | DINO (ResNet-50) | ImageNet [†] | ImageNet [†] | 68.4 (512) | ↑ **1.3%** | ↑ **0.9%** |
| | | DINO (ViT-S/16) | ImageNet [†] | ImageNet [†] | 74.8 (256) | ↑ **0.6%** | ↑ **0.3%**[‡] |
| Argument Retrieval | ArguAna (*Rec@100*) | ANCE (BERT) | MSMarco | WT20[†] | 94.5 (64) | ↑ **0.8%** | ↑ **0.5%**[§] |

off-the-shelf Bert-Siamese model called ANCE[4] (Xiong et al., 2021) trained for document retrieval, following the dataset definitions from Thakur et al. (2021).

We do not use any supervision for learning TLDR nor any of the methods we compare to. Given a trained dimensionality reduction encoder, we then encode all downstream task data for each test dataset, and evaluate them in a "zero-shot" manner, using non-parametric classifiers (*k*-NN) for all retrieval tasks. Specifically, for landmark image retrieval on $\mathcal{R}$Oxford/$\mathcal{R}$Paris we use the common protocols presented by Radenović et al. (2018a), where specific queries are defined. For every query, we measure the mean average precision metric across all other images depicting the same landmark. For ImageNet Russakovsky et al. (2015), we follow the exact process used by Caron et al. (2021) and others: The gallery is composed of the full validation set, spanning 1000 classes, and each image from this validation (val) set is used in turn as a query. Once again, we use *k*-NN to compute a list of images ranked by their relevance to the query, and we aggregate the labels from the top 20 images, assigning the predicted query label to the most prominent class. In Table 1 we report a summary of the most notable results in each case. We chose $d = 128$ (resp. 64) for the visual (resp. textual) domains as the most commonly used setting in practice.

**Implementation details.** We do not explicitly normalize representations during learning TLDR; yet, we follow the common protocol and L2-normalize the features before retrieval for both tasks. Results reported for PCA use whitened PCA; we tested multiple whitening power values and kept the ones that performed best. Further implementation details are reported in the Appendix. It is noteworthy that we used the *exact same hyper-parameters* for the learning rate, weight decay, scaling, and $\lambda$ suggested in Zbontar et al. (2021), despite having very different tasks and encoder architectures. We were able to run PCA on up to millions of data points and hundreds of dimensions using out-of-the-box tools. More precisely, we use the PCA implementation from scikit-learn.[5] For large matrices ($> 500\text{x}500$, which is our case) it uses the randomized SVD approach by Halko et al. (2011), which is an approximation of the full SVD. This has been the standard way of scaling PCA to large matrices.

**Measuring variance.** The proposed method has some inherent stochasticity, e.g. from SGD or from the neighbor pair sampling step (Step 2 in Algorithm 1). To make sure we properly measure variance, we run each variant of TLDR shown in Figures 2 and 6 for five times and average the output results; the error bars report the standard deviation across those 5 runs. The reason we only report error bars for TLDR is

---

[4]https://www.sbert.net/docs/pretrained_models.html
[5]https://scikit-learn.org/stable/modules/generated/sklearn.decomposition.PCA.html

Table 2: **Compared Methods.** For *unsupervised* methods, the objective is based on reconstruction. *Neighbor-supervised* methods use nearest neighbors as pseudo-labels to guide the learning. *Denoising* learns to ignore added Gaussian noise. Note that in Figure 6a, we compare to a few additional manifold learning methods (*i.e.* ICA, LLE, UMAP, LPP and LTSA) that could not scale to large output dimensions.

| Method | (Self-) supervision | Encoder | Projector | Loss | Notes |
|---|---|---|---|---|---|
| PCA (Pearson, 1901) | unsupervised | linear | linear | Reconstruction MSE + orthogonality | Used for dimensionality reduction in SoTA methods like DELF, GeM, GeM-AP and HOW |
| DrLim Contrastive | neighbor-supervised neighbor-supervised | MLP linear | None MLP | Contrastive Contrastive | (Hadsell et al., 2006) (*very low performance*) Hadsell et al. (2006) with projector |
| MSE $TLDR_{\mathcal{G}}$ | unsupervised denoising | linear linear | MLP MLP | Reconstruction MSE Barlow Twins | TLDR with MSE loss TLDR with noise as distortion |
| TLDR $TLDR_{1,2}$ $TLDR_{1,2}^{\star}$ | neighbor-supervised neighbor-supervised neighbor-supervised | linear fact. linear MLP | MLP MLP MLP | Barlow Twins Barlow Twins Barlow Twins | |

because it has some noticeable variance; all other methods either correspond to deterministic algorithms or had negligible variance and therefore the error bars are not visible.

**Further ablations, results on FashionMNIST and 2D visualizations.** Additional results and interesting ablations can be found in the Appendix. In particular, we explore the effect of the training set size, the batch size and report results on another NLP task: duplicate query retrieval. Moreover, and although beyond the scope of what TLDR is designed for, in Appendix D we present results on FashionMNIST when using TLDR on raw pixel data and for 2D visualization. We show that for cases where the input pixels are forming an informative space, TLDR can achieve top performance for $d \geq 8$.

### 3.2 Results on image retrieval

We first focus on landmark image retrieval. For large-scale experiments on this task, it is common practice to apply dimensionality reduction to global normalized image representations using PCA with whitening (Jégou & Chum, 2012; Tolias et al., 2016; Revaud et al., 2019; Tolias et al., 2020). For the experiments in this section, we simply replace the standard PCA step with our proposed TLDR. We conduct most comparisons and ablations using the optimized, retrieval-oriented GeM-AP representations proposed by Revaud et al. (2019) that are tailored to the landmark image retrieval task and present results in Section 3.2.1. We then report results using the generic DINO representations that are learned in a self-supervised manner in Section 3.2.2.

#### 3.2.1 Landmark image retrieval using GeM-AP representations

**Experimental protocol.** We start from 2048-dimensional features obtained from the pre-trained ResNet-50 of (Revaud et al., 2019), which uses Generalized-Mean pooling (Radenović et al., 2018b) and has been specifically trained for landmark retrieval using the AP loss (GeM-AP). To learn the dimensionality reduction function, we use GLD-v2, a dataset composed of 1.5 million landmark images (Weyand et al., 2020). We learn different output spaces whose dimensions range from 32 to 512. We evaluate these spaces on two standard image retrieval benchmarks (Radenović et al., 2018a), the revisited Oxford and Paris datasets ($\mathcal{R}$Oxford and $\mathcal{R}$Paris). Each dataset comes with two test sets of increasing difficulty, the "Medium" and "Hard". Following these datasets' protocol, we apply the learned dimensionality reduction function to encode both the gallery images and the set of query images whose 2048-dim features have been extracted beforehand with the model of Revaud et al. (2019). We then evaluate landmark image retrieval on ROxford5K and RParis6K and report mean average precision (mAP), the standard metric reported for these datasets. For brevity, we report the "Mean" mAP metric, *i.e.* the average of the mAP of the "Medium" and "Hard" test sets; we include the individual plots for "Medium" and "Hard" in the Appendix for completeness.

**Compared approaches.** We report results for several flavors of our approach. *TLDR* uses a linear projector, $TLDR_1$ uses a factorized linear one, and $TLDR_1^{\star}$ an MLP encoder with 1 hidden layer. As an alternative, we

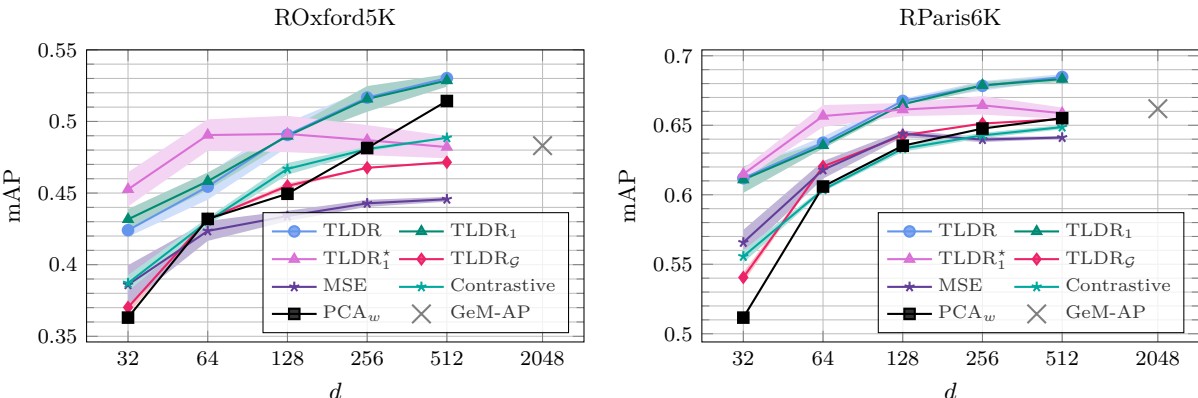

Figure 2: **Landmark image retrieval experiments using GeM-AP representations**. Mean average precision (mAP) on $\mathcal{R}$Oxford (left) and $\mathcal{R}$Paris (right) as a function of the output dimensions $d$. We report TLDR with different encoders: linear (TLDR), factorized linear with 1 hidden layer (TLDR$_1$), and a MLP with 1 hidden layer (TLDR$_1^\star$), the projector remains the same (MLP with 2 hidden layers). We compare with PCA with whitening, two baselines based on TLDR, but which respectively train with a reconstruction (MSE) and a contrastive (Contrastive) loss, and also with $TLDR_\mathcal{G}$, a variant of TLDR which uses Gaussian noise to synthesize pairs. The original GeM-AP performance is also reported.

also report $TLDR_\mathcal{G}$, which uses Gaussian noise to create synthetic neighbour pairs. All variants use an MLP with 2 hidden layers and 8192 dimensions as a projector.

We compare with a number of un- and self-supervised methods (see also Table 2 for a summary). First, and foremost, we compare to reducing the dimension with *PCA* with whitening, which is still standard practice for these datasets (Revaud et al., 2019; Radenović et al., 2018b; Tolias et al., 2020). We also report results for our approach but trained with the Mean Square Error reconstruction loss instead of the Barlow Twins' (as we discuss in Section 4, PCA can be rewritten as learning a linear encoder and projector with a reconstruction loss), and refer to this method as *MSE*. In this case, the projector's output is reduced to 2048 dimensions in order to match the input's dimensionality. Following a number of approaches that use nearest neighbors as (self-)supervision for contrastive learning (Hadsell et al., 2006), the *Contrastive* approach uses a contrastive loss on top of the projector's output. This draws inspiration from Hadsell et al. (2006), and is a variant where we replace the Barlow Twins loss, with the loss from Hadsell et al. (2006). It is worth noting that we omit results from a more faithful reimplementation of Hadsell et al. (2006), *i.e.* using a max-margin loss directly applied on the lower dimensional space and without a projector, as they were very low. Note that we considered other manifold learning methods, but none of the ones we tested was able to neither scale, nor outperform PCA in output dimensions $d \geq 8$; we present comparisons for smaller $d$ in Section 3.4. Finally, we report retrieval results obtained on the initial features from Revaud et al. (2019) (*GeM-AP*), *i.e.* without dimensionality reduction. For all flavours of TLDR, we fix the number of nearest neighbors to $k = 3$, although, and as we show in Figure C, TLDR performs well for a wide range of number of neighbors.

**Results.** Figure 2 reports mean average precision (mAP) results for $\mathcal{R}$Oxford and $\mathcal{R}$Paris; as the output dimensions $d$ varies. We report the average of the Medium and Hard protocols for brevity, while results per protocol are presented in Appendix B.1. We make a number of observations. First and most importantly, we observe that both linear flavors of our approach outperform PCA by a significant margin. For instance, TLDR improves ROxford5K retrieval by almost 4 mAP points for 128 dimensions over the PCA baseline. The MLP flavor is very competitive for very small dimensions (up to 128) but degrades for larger ones. Even for the former, it is not worth the extra-computational cost. An important observation is that we are able to retain the performance of the input representation (GeM-AP) while using only 1/16th of its dimensionality. Using a different loss (MSE and Contrastive) instead of the Barlow Twins' in TLDR degrades the results.

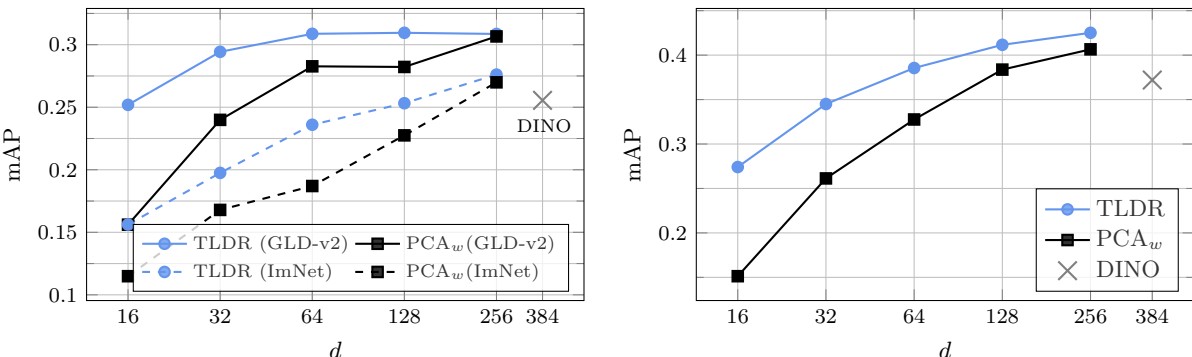

Figure 3: **Self-supervised landmark retrieval performance on $\mathcal{R}$Oxford using DINO ViT-S/16 backbones**. Left: mAP on $\mathcal{R}$Oxford as a function of the output dimensions $d$ using representations from DINO pretrained on ImageNet; dimensionality reduction is learned on either ImageNet (dashed lines) or GLD-v2 (solid lines). Right: Performance for TLDR and PCA when learning dimensionality reduction on GLD-v2 over representations from a DINO model pretrained on GLD-v2. No labels are used at any stage.

These approaches are comparable to or worse than PCA. Finally, replacing true neighbors by synthetic ones, as in TLDR$_\mathcal{G}$, performs worse.

### 3.2.2 Landmark image retrieval using DINO representations

In the previous section, we started from representations pretrained using supervision and tailored to the landmark retrieval task; we then learned dimensionality reduction on top of those in an unsupervised way. It is however interesting to also study the *fully unsupervised case* and measure how well the method performs if we use representations learned in a self-supervised way. In this section, we assume that images are represented using DINO (Caron et al., 2021), a state-of-the-art self-supervised approach that whose representations have lead to impressive results for a wide variety of transfer learning tasks, including image retrieval.

**Results.** In Figure 3 we report results on $\mathcal{R}$Oxford when learning dimensionality reduction on top of DINO features from a ViT-S/16 backbone. In all cases, we follow the evaluation protocol presented in the previous section. In Figure 3 (left) we present results starting from the publicly available ViT DINO model trained on ImageNet; similar to the GeM-AP case, we treat the ViT as a feature extractor and learn a linear encoder on top using either the GLD-v2 or ImageNet datasets. In all cases, both features and dimensionality reduction are learned without any supervision. We see that TLDR shows strong gains over PCA with whitening, the best performing competing method, and that gains are consistent across multiple values of output dimension $d$ and across all setups. In fact, we see that if one assumes access to unlabeled data from the downstream landamark domain, one can achieve a large **+5.4 mAP gain** over DINO, **by simply learning a linear encoder on top**, and **without the need to fine-tune the ViT model**. It is also noteworthy that TLDR is able to **match the DINO ViT performance** on $\mathcal{R}$Oxford **using only 16-dimensions**. Results for $\mathcal{R}$Paris are presented in the appendix.

In Figure 3 (right) we start from a publicly available DINO model trained in an unsupervised way on GLD-v2. Caron et al. (2021) evaluate their representations on $\mathcal{R}$Oxford and $\mathcal{R}$Paris using global image features from this model; they report 37.9% mAP on $\mathcal{R}$Oxford for the average setting (0.52/0.24 for medium/hard splits). We see that TLDR can improve on that result and is able to achieve **state-of-the-art** performance on $\mathcal{R}$Oxford when using self-supervised learning, *i.e.* 0.43 mAP (0.57/0.28 for medium/hard splits) for $d = 256$, **+4.6% mAP higher** than using the original DINO features learned on GLD-v2.

### 3.2.3 Results on ImageNet

We further evaluate the performance of the proposed TLDR on ImageNet retrieval using $k$-NN. We follow the protocol from Caron et al. (2021); Wu et al. (2018) and run the corresponding evaluation scripts provided in the DINO codebase. We query with all images in the ImageNet val set, using the training set as the database;

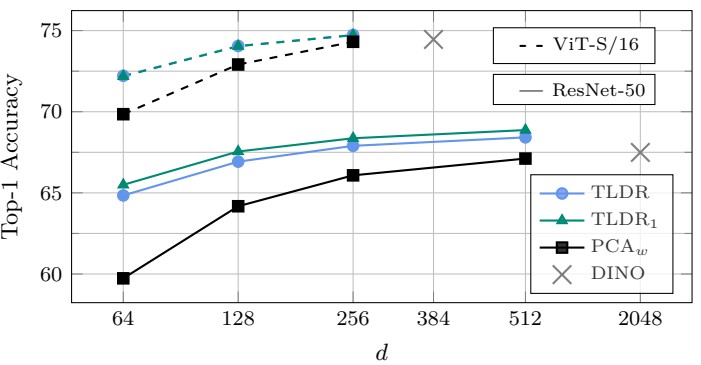 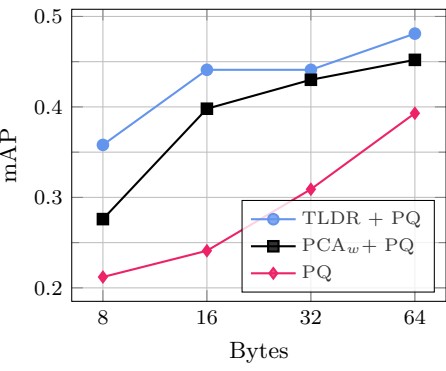

(a) $k$-NN Accuracy on ImageNet using unsupervised learning.

(b) $\mathcal{R}$Oxford mAP after quantization.

Figure 4: **Retrieval results on ImageNet, and after vector quantization.** Left: Top-1 Accuracy as a function of the output dimensions $d$ for $k$-NN retrieval on ImageNet following the protocol from Caron et al. (2021) and using DINO ResNet-50 and ViT representations trained on ImageNet. Right: Performance on $\mathcal{R}$Oxford after PQ quantization of the reduced features ($d = 128$), as a function of the output vector size.

similar to DINO, we then report results using 20-NN. We report Top-1 Accuracy in Figure 4a for the DINO ResNet-50 and ViT-S/16 models. We see that TLDR is consistently better than PCA also in this case, and for ResNet-50 gives +9.4/5.1/2.7% gains over PCA for $d$=32/64/128. For ViT, performance is higher and gains smaller, but still consistent. Generally, linear TLDR encoders (linear and factorized linear) **improve DINO's retrieval performance** on ImageNet $k$-NN for both backbones and for all output dimension over $d = 128$; this is not the case for PCA, which is not able to improve DINO's performance. We further see that TLDR is able to outperform the original 2048-dimensional features with only 256 dimensions, *i.e.* **achieving a $10\times$ compression without loss in performance** for ResNet-50, or reach approximately 75% Top-1 Accuracy on ImageNet with 256-dimensional features for ViT-S/16 DINO.

### 3.3   Results on first stage document retrieval

For document retrieval, the process is generally divided into two stages: the first one selects a small set of candidates while the second one re-ranks them. Because it works on a smaller set, this second stage can afford costly strategies, but the first stage has to scale. The typical way to do this is to reduce the dimension of the representations used in the first retrieval stage, often in a supervised fashion (Khattab & Zaharia, 2020; Gao et al., 2021). Following our initial motivation, we investigate the use of unsupervised dimensionality reduction for document retrieval scenarios where a supervised approach is not possible.

**Experimental protocol.** We start from 768-dimensional features extracted from a model trained for Question Answering (QA), *i.e.* ANCE (Xiong et al., 2021). We use Webis-Touché-2020 (Bondarenko et al., 2020; Wachsmuth et al., 2017) a conversational argument dataset composed of 380k documents to learn the dimensionality reduction function. **Compared approaches.** We report results for three flavors of our approach. *TLDR* uses a linear encoder while $TLDR_1$ and $TLDR_2$ use a factorized linear one with respectively one hidden layer and two hidden layers. We compare with PCA, which was the best performing competitor from Section 3.2. We also report retrieval results obtained with the 768-dimensional initial features.

**Results.** Figure 5 reports retrieval results on ArguAna, for different output dimensions $d$. We observe that the linear version of TLDR outperforms PCA for almost all values of $d$. The linear-factorized ones outperforms PCA in all scenarios. We see that the gain brought by TLDR over PCA increases as $d$ decreases.

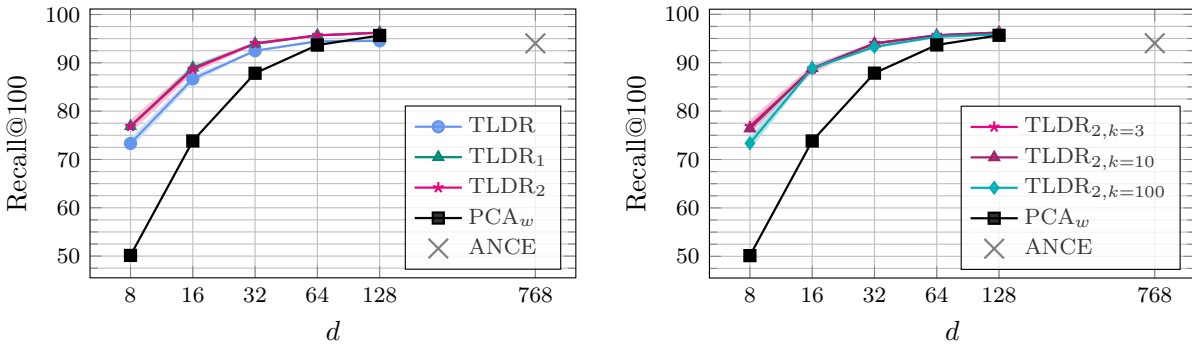

Figure 5: **Argument retrieval results on ArguAna** for different values of output dimensions $d$. On the left we vary the amount of factorized layers, with fixed $k = 3$, on the right we fix the amount of factorized layers to 2 and test $k = [3, 10, 100]$. Factorized linear is fixed to 512 hidden dimensions.

Note that we achieve results equivalent to the initial ANCE representation using only 4% of the original dimensions; PCA, needs twice as many dimensions to achieve similar performance.

### 3.4 Analysis and Impact of hyper-parameters

**Impact of hyper-parameters.** We observed surprising robustness for TLDR across multiple hyper-parameters. For brevity, we merely summarize the main findings here, while complete figures are presented in the appendix. When we vary the **architecture of the projector** $g_\phi$, an important module of TLDR, we see that having hidden layers generally helps. As also noted in Zbontar et al. (2021), having a high auxiliary dimension $d'$ for computing the loss is very important and highly impacts performance. When it comes to the **numbers of neighbors** $k$, we see that TLDR is surprisingly consistent across a wide range of $k$. We observe the same stability across several **batch sizes**. More details are presented in the appendix.

**Comparisons to manifold learning methods on smaller output dimensions.** In Figure 6a we present results for TLDR when the output dimensionality is $d \leq 32$; in this regime, a few more manifold learning methods can be run, *e.g.* ICA, Locally Linear Embedding (LLE) (Roweis & Saul, 2000), Local Tangent Space Alignment (LTSA) (Zhang & Zha, 2004), Locality Preserving Projections (LPP) (He & Niyogi, 2003) and UMAP (McInnes et al., 2018). Unfortunately, even at smaller output dimensions we had to subsample the dataset to run some of the methods, due to their scalability issues. Specifically, we are forced to use only 5% of the training set ($\sim$75K images) for learning LLE and LTSA, 10% for LPP ($\sim$150K - this setting gives the best results) and 50% ($\sim$750K images) for UMAP. We see that TLDR generally dominates for $d \geq 8$, and that linear methods tend to be the highest performing ones as the output dimensionality grows.

**How sensitive are TLDR-reduced vectors to subsequent quantization?** In Figure 4b we verify that the gains from TLDR persist after compressing the dimensionality reduced vectors to only a few bytes, *e.g.* via Product Quantization (PQ) (Jegou et al., 2010). We conduct the study on $\mathcal{R}$Oxford, starting from the GeM-AP ResNet-50 representations. First of all, we see that it is highly desirable to first use dimensionality reduction and then PQ-compress the vectors; starting from the 2048-dimensional vectors results to much lower performance. Secondly, we see that TLDR once again highly outperforms PCA, with +8.16% and +4.28% in mAP for compression to 64 or 128 *bits*, respectively.

**How sensitive is TLDR to approximate nearest neighbors?** To verify that our system is robust to an approximate computation of nearest neighbors, we test its performance using product quantization (Jegou et al., 2010; Ge et al., 2013) while varying the quantization budget (i.e. the amount of bytes used for each image during the nearest neighbor search). Compression is done using optimized product quantization (OPQ) (Ge et al., 2013) via the FAISS library (Johnson et al., 2017) and results are reported in Figure 6b. We present results on $\mathcal{R}$Oxford, starting from the GeM-AP ResNet-50 representations. We see that TLDR

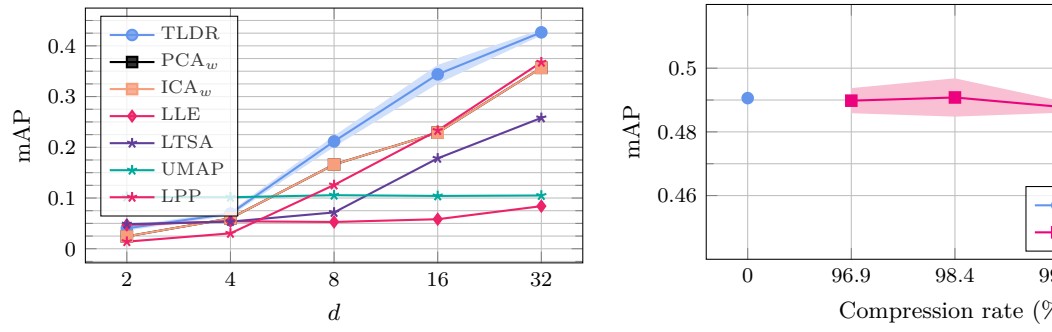

(a) Comparisons to manifold learning methods.

(b) Effect of approximate nearest neighbors.

Figure 6: Left: Comparisons to manifold learning methods for small output dimensions $d \leq 128$; we report Mean average precision (mAP) on $\mathcal{R}$Oxford using GeM-AP representations as a function of the output dimensions $d$. Right: The effect of nearest neighbor approximation for $d = 128$. We plot mAP as a function of the compression rate used during nearest neighbor computation. Note that the baseline (compression rate = 0) is using the 2048-dimensional (8192 bytes) GeM-AP representations during nearest neighbor computation.

is quite robust to quantization during the nearest neighbor search and that even when the quantization is pretty strong (1/64 the default size or merely 16 bytes per vector) TLDR still retains its high performance.

## 4 Related work

The basic idea behind TLDR is embarrassingly simple and links to a large number of related methods, from PCA to manifold learning and neighborhood embedding. In this section we discuss a few such relations; more are discussed in Appendix E.

**Linear dimensionality reduction.** We refer the reader to Cunningham & Ghahramani (2015) for an extensive review of linear dimensionality reduction. It is beyond the scope of this paper to exhaustively discuss many such related works, we will therefore focus on PCA (Pearson, 1901) which is the de facto standard linear dimensionality method, in particular for large-scale retrieval.

One can derive the learning objective of PCA (Pearson, 1901) by setting $f_\theta(x) = W^T x$ and $g(x) = Wx$ in the model of Figure 1, *i.e.* use a linear encoder and projector with $W \in \mathrm{R}^{D \times d}$, and optimize $W$ via minimizing the Frobernius norm of the matrix of reconstruction errors over the whole training set, subject to orthogonality constraints:

$$W^* = \arg \min_{W} ||x - g(f_\theta(x))||_F, \quad \text{s.t.} \quad W^T W = I_d. \tag{2}$$

This equation has a closed form solution that can be obtained via the eigendecomposition of the data covariance matrix and then keeping the largest $d$ eigenvectors[6].

Unlike PCA, TLDR does not constrain the projector to be a linear model, nor the loss to be a reconstruction loss. In fact, the redundancy reduction term in the Barlow Twins loss encourages the *whitening* of the batch representations as a soft constraint (Zbontar et al., 2021), in a way analogous to the orthogonality constraint of Eq.(2). We see from Figure C that part of the performance gains of TLDR over PCA is precisely due to this asymmetry in the architecture, *i.e.* when the projector is an MLP with hidden layers. Looking at MSE results in Figures 2, *i.e.* a version of TLDR with a reconstruction loss, we also see that the Barlow Twins loss and the flexibility of computing it in an arbitrarily high $d'$-dimensional space further contributes to this gain. One can therefore interpret TLDR *as a more generic way of optimizing a linear encoder,* i.e. *using an arbitrary decoder and approximating the constraint of Eq.(2) in a soft way,* further incorporating a weak notion of whitening.

---

[6]We refer the reader to Chapter 2 of the Deep Learning book (Goodfellow et al., 2016) for the derivation of the PCA solution.

Another way of learning a linear encoder is presented in He & Niyogi (2003) where Locality Preserving Projections are given by the optimal linear approximations to the eigenfunctions of the Laplace-Beltrami operator on the data manifold. We find the TLDR optimization to not only be more scalable, but also more versatile as it is stochastically and independently optimizing for different neighborhoods of the manifold.

**Manifold learning and neighborhood embedding methods.** Manifold learning methods define objective functions that try to preserve the local structure of the input manifold, usually expressed via a $k$-NN graph. Non-linear unsupervised dimensionality-reduction methods usually require the $k$-NN graph of the input data, while most further require eigenvalue decompositions (Roweis & Saul, 2000; Donoho & Grimes, 2003; Zhang & Zha, 2004) and shortest-path (Tenenbaum et al., 2000) or computation of the graph Laplacian (Belkin & Niyogi, 2003). Others involve more complex optimization (McInnes et al., 2018; Agrawal et al., 2021). Moreover, many manifold learning methods were created to solely operate on the data they were learned on. Although "out-of-sample" extensions for many of such methods have been proposed (Bengio et al., 2004), methods like Spectral Embeddings, pyMDE (Agrawal et al., 2021) or the very popular $t$-SNE (Van der Maaten & Hinton, 2008) can only be used for the data they were trained on. Finally, UMAP (McInnes et al., 2018) was recently proposed as not only a competitor of $t$-SNE on 2-dimensional outputs, but as a general purpose dimension reduction technique. Yet, all our experiments with UMAP, even after exhaustive hyperparameter tuning, resulted in very low performance for $d \geq 8$ for all the tasks we evaluated.

**Nearest neighbors as "supervision" for contrastive learning.** The seminal method DrLIM (Hadsell et al., 2006) uses a contrastive loss over neighborhood pairs for representation learning. Experimenting only on simple datasets like MNIST, it learns a CNN backbone and the dimensionality reduction function in a single stage, using a max-margin loss. TLDR resembles DrLIM (Hadsell et al., 2006) with respect to the encoder input and the way pairs are constructed; a crucial difference, however, is the loss function and the space in which it is computed: DrLIM uses a contrastive loss which is computed directly on the lower dimensional space. Despite our best effort to make this approach work as described, performance was very low without a projector. Using the contrastive loss from Hadsell et al. (2006), together with the projector we use for TLDR, we were able to get more meaningful results (reported as Contrastive in our experiments), although still underperforming the Barlow Twins loss. This difference may be due to two reasons: first, and as discussed above, Barlow Twins encourages the whitening of the representations which makes it more suitable for this task. Second, and like many other pair-wise losses, the contrastive loss further requires sampling hard/meaningful negatives (Wu et al., 2017; Radenović et al., 2018b). Very recently, Iscen et al. (2022) use features space neighbor consistency as a regularization term for learning under label noise by encouraging the prediction of each example to be similar to its nearest neighbours'.

**Relation to Deep Canonical Correlation Analysis.** The MLP variant of TLDR is closely related to the Deep Canonical Correlation Analysis (DCCA) introduced by Andrew et al. (2013). The Barlow Twins loss can be seen as a more modern, simpler way to optimize DCCA. First, the addition of the projector is shown to significantly and positively impact the results. Second, the gradient of DCCA involves computing the gradient of the trace of the square-root of a matrix, a step which is not required for computing the Barlow Twins loss. Moreover, Deep CCA obtains optimal results with full-batch optimization and L-BFGS, *i.e.* a far less scalable setting.

**Temporal neighborhoods.** Component analysis methods that take into account the temporal structure of the data are also related, something critical for time series. Dynamical Components Analysis (DCA) (Bai et al., 2020), for example, uses *temporal* neighborhoods, to discover a subspace with maximal predictive information, defined as the mutual information between the past and future. Similarly, Deep Autoencoding Predictive Components (DAPC) (Clark et al., 2019) builds on the same notion of predictive mutual information from temporal neighbors, but further adopts a masked reconstruction task to enforce the latent representations to be more informative.

**Graph diffusion for harder pairs.** Iscen et al. (2018b) improve the method from Hadsell et al. (2006) by mining harder positives and negative pairs for the contrastive loss via diffusion over the $k$-NN graph. Similar to Hadsell et al. (2006), they learn (fine-tune) the whole network and not just the dimensionality-reduction layer. Although it would be interesting to incorporate such ideas in TLDR, we consider it complementary and beyond the scope of this paper. Methods for learning descriptor matching are also related; *e.g.* (Simonyan et al., 2014) formulates dimensionality reduction as a convex optimisation problem. Although the redundancy

reduction objective can be formulated in many ways, *e.g.* via stochastic proximal gradient methods like Regularised Dual Averaging in Simonyan et al. (2014), we believe that the *simplicity, immediacy and clarity* in which the Barlow Twins objective optimizes the output space is a strong advantage of TLDR.

## 5    Discussion

**The Barlow Twins and related SSL losses.** We chose the Barlow Twins loss from Zbontar et al. (2021) as we believe it fits dimensionality reduction exceptionally. The second term of the loss does not only provide Barlow Twins with an elegant way of avoiding collapse, something not trivial for other related losses like SimSiam (Chen & He, 2020; Zhang et al., 2022), but decorrelating the output space is a property that highly suits the dimensionality reduction task. Moreover, from our experiments and ablations, we see that the Barlow Twins loss is highly robust to most hyper-parameters. We could however devise a variant of TLDR that uses any other "symmetrical" loss, *e.g.* SimCLR (Chen et al., 2020a), SimSiam (Chen & He, 2020) or the recently proposed VICReg (Bardes et al., 2022) that builds on Barlow Twins. Yet, leveraging SSL losses that use self-distillation and the Exponential Moving Average trick such as BYOL (Grill et al., 2020) or MoCo (He et al., 2020) with neighbor pairs might be far from trivial.

**TLDR beyond dimensionality reduction.** As we discuss in the previous paragraph, we do not consider TLDR to be a method for learning large visual representation models from scratch, where neighbors would simply replace image transformations. We however believe that this signal can be useful as another way of performing model distillation, by *e.g.* sampling neighbors from a larger model or a model trained on more data. It is also clear from the results of TLDR learning on top of the generic ImageNet-trained DINO and the GLD-v2 dataset in Figure 3 (left), that our method can be used for unsupervised domain adaptation. TLDR is also agnostic to the way neighbors are extracted, and other similarity measures could be used. The choice of this measure would then be a hyper-parameter, potentially taking prior information into account. TLDR could then learn an (output) embedding space for which the Euclidean distance is suitable, a highly desirable property for many tasks.

**A scalable manifold learning method.** By "scalability" we refer to the ability of the proposed TLDR to be efficiently applied to arbitrary large datasets and output spaces. We believe this to be an important property, given that most manifold learning methods are unable to "scale" to large datasets or output dimensions. This is why, in Figure 6a for example, we had to subsample the training dataset so some of the competing methods could fit in memory and/or converge within a couple of days (see also Appendix B.6 for training time comparisons). TLDR uses mini-batch Stochastic Gradient Descent and has linear time and memory complexity with respect to both the training database size and the output dimension. In that regard, this is a highly scalable manifold learning method.

**Computational cost and training time.** In terms of computation, it is worth noting that training TLDR, *i.e.* learning the dimensionality reduction encoder, is generally orders of magnitude less expensive than than *e.g.* learning visual representation. We always start from pre-extracted feature vectors and learn a small (linear or MLP) encoder and an MLP projector. There is no large backbone to learn and even datasets like GLD-v2 (Weyand et al., 2020) (composed of 1.5M vectors in 2048 dimensions for the GeM-AP model) can easily fit in memory. When it comes to training time, TLDR is noticeably slower than PCA. For example, for $d = 128$ and for the GLD-v2 dataset, learning PCA takes approximately 18 minutes (multi-core), while learning a linear TLDR over 100 epochs takes approximately 63 minutes (on a single GPU). The latter includes 13 minutes for computing (exact) k-NNs for the dataset.

Although slower at training time, linear TLDR highly outperforms PCA, while all other manifold learning methods that we compare to can neither reach the performance of PCA nor be applied to large training datasets for $d \geq 32$. More importantly, as we show in Appendix B.4, the performance of PCA saturates early and, unlike TLDR, does not benefit from adding more data; Our study on a dataset of 1.5 million images suggests that the gap with PCA increases as the training set gets larger. Finally, although the training time of TLDR is higher than PCA, both methods share the same *linear encoding complexity* during testing, something highly important as this is a process repeated at every single retrieval request.

**TLDR out of its comfort zone.** Although TLDR can be seen as a way of generalizing recent self-supervised visual representation learning methods to cases where handcrafted transformations of the data are challenging

or impossible to define, we want to emphasize that it *is not suited* for self-supervised representation learning from pixels; augmentation invariance is a much more suited prior in that regard, while it is also practically impossible to define meaningful neighboring pairs from the input pixel space. Additionally, although visualization is a common manifold learning application, TLDR is neither designed not recommended for 2D outputs; there are other methods like *t*-SNE, UMAP or MDE that specialize for such tasks (see also Appendix).

**What is TLDR suitable for?** TLDR excels for dimensionality reduction to mid-size outputs, *e.g.* when $d$ is between 32 and 256 dimensions, a range to which the vast majority of manifold learning methods cannot scale. This is very useful in practice for retrieval. TLDR further provides a computationally efficient way of *adapting* pre-trained representations, coming *e.g.* from large pre-trained models, to a new domain or a new task, without the need for any labeled data from the downstream task, and without fine-tuning large encoders.

## 6    Conclusions

In this paper we introduce TLDR, a dimensionality-reduction method that combines neighborhood embedding learning with the simplicity and effectiveness of recent self-supervised learning losses.

By simply replacing PCA with TLDR, one can significantly increase the performance of generic (DINO) or specialized (GeM-AP) models on retrieval tasks like ImageNet or $\mathcal{R}$Oxford, especially if one has access to *unlabeled* data from the downstream retrieval domain. Those gains are achieved by learning a linear encoder on top of pre-extracted features and without the need to fine-tune a large backbone. To summarize, TLDR offers a number of desirable properties:

i) *Scalability:* learned via stochastic gradient descent, TLDR can easily be parallelized across GPUs and machines, while for even the largest datasets, approximate nearest neighbor methods can be used to create input pairs in sub-linear complexity (Babenko & Lempitsky, 2014; Kalantidis & Avrithis, 2014), ii) *Simplicity:* The Barlow Twins (Zbontar et al., 2021) objective is robust and easy to optimize, and does not have trivial solutions, iii) *Out-of-sample generalization*, and iv) *Linear encoding complexity:* TLDR is highly effective with a linear encoder, offering a direct replacement of PCA without extra encoding cost.

## Acknowledgements

The authors want to sincerely thank Christopher Dance for his early comments that helped shape this work. We also want to thank our colleagues at NAVER LABS Europe[7] for providing feedback that made this work significantly better. Specifically we would like to thank Gabriela, Jérome, Boris, Martin, Stéphane, Bulent, Rafael, Gregory, Riccardo and Florent for reviewing our work and providing us with constructive comments. Last but not least, we want to thank Audi for hand-drawing our gorgeous teaser Figure. Finally, we want to thank all anonymous reviewers, whose comments also made this work better. This work was supported in part by MIAI@Grenoble Alpes (ANR-19-P3IA-0003).

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

# Appendix

## A    Appendix Summary

In this appendix we present a number of additional details, results and Figures that we could not fit in the main text:

- We report additional experiments for the **landmark retrieval** task in Appendix B. Specifically, we present results for the med/hard splits separately for GeM-AP (Appendix B.1), an experiment with oracle neighbors (Appendix B.2), an experiment with features from a larger ResNet-101 backbone (Appendix B.7). We also present some additional results with DINO representations (Appendix B.8).

- We present ablations when we **vary the projector architecture** and the **number of neighbors** $k$ (Appendix B.3), as well as when **varying the training set size** (Appendix B.4) and the **batch size** (Appendix B.5). We also compare the training times of TLDR and other manifold learning methods as we vary the size of the training set (Appendix B.6)

- We report additional experiments for the **document retrieval** in Appendix C. Specifically, we extend our evaluation protocol and report result on a new task: **duplicate query retrieval**. We further investigate not only dimensionality reduction for the same task, but also the case of **dimensionality reduction transfer**.

- Although TLDR is not suited for such applications, as a proof of concept we present results on the FashionMNIST dataset in Appendix D, *i.e.* when **learning from raw pixel data**. We also present some results when using TLDR for **visualization**, *i.e.* when the output dimension is $d = 2$ in Figure N.

- We extend Section 4 with **further discussion** on related topics and more related works in Appendix E. We conclude with a brief discussion on limitations of TLDR in Appendix E.1

- Finally, in Appendix F we give pytorch-style pseudocode for initializing and training TLDR.

## B    Additional experiments: Landmark image retrieval

### B.1    Results on Medium and Hard protocols separately

In Figure A we report the mAP metric for the Medium and Hard splits of the Revisited Oxford and Paris datasets (Radenović et al., 2018a) separately.

### B.2    Results with "Oracle" nearest neighbors

In Figure B we present results using an oracle version of TLDR, *i.e.* a version that uses labels to only keep as pairs neighbors that come from the same landmark in the training set. As we see, TLDR practically matches the oracle's performance.

### B.3    Varying the projector architectures and the number of neighbors

Figure C studies the role of some of our parameters. On the left size of Figure C, we vary the architecture of the projector $g_\phi$, an important module of TLDR. We see that having hidden layers generally helps. As also noted in Zbontar et al. (2021), having a high auxiliary dimension $d'$ for computing the loss is very important and highly impacts performance. On the right side of Figure C we show the surprisingly consistent performance of TLDR across a wide range of numbers of neighbors $k$.

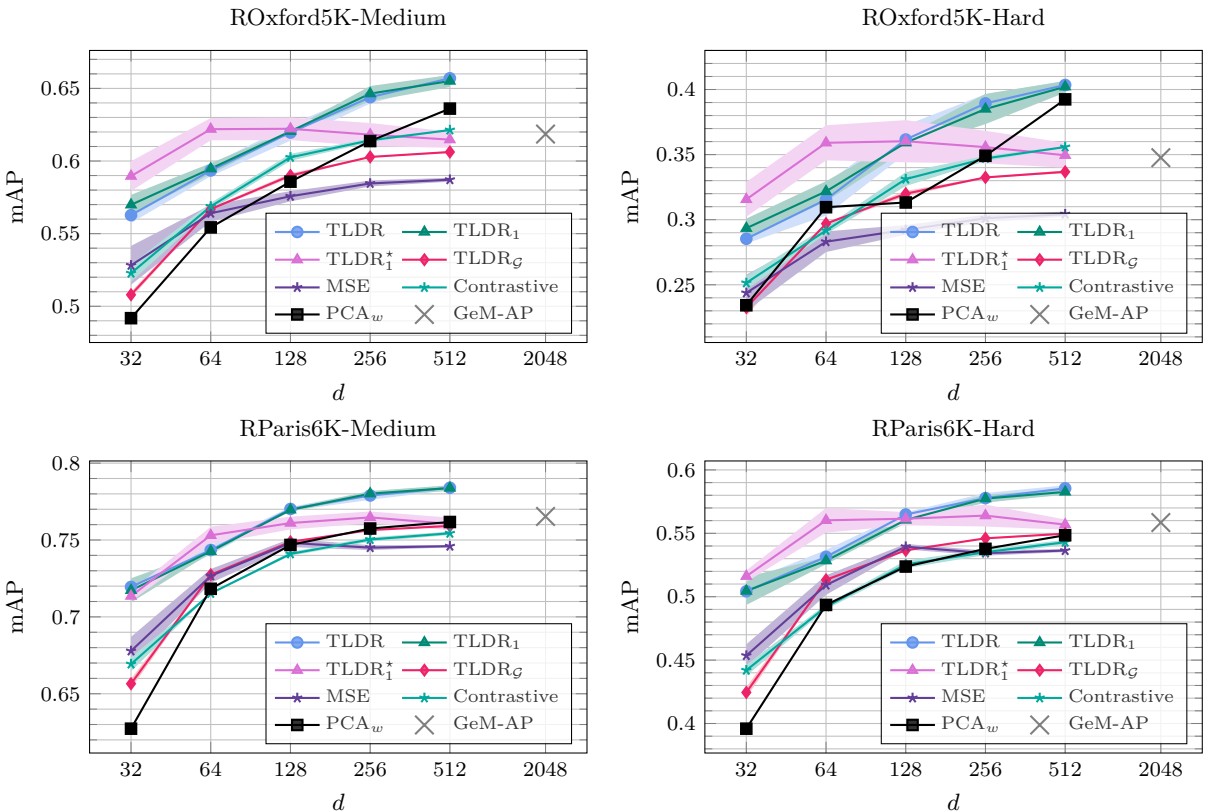

Figure A: **Image retrieval experiments**. Mean average precision (mAP) on $\mathcal{R}$Oxford (top) and $\mathcal{R}$Paris (bottom), for the Medium (left) and Hard (right) test sets, as a function of the output dimensions $d$. We report TLDR with different encoders: linear (TLDR), factorized linear with 1 hidden layer (TLDR$_1$), and a MLP with 1 hidden layer (TLDR$_1^\star$), the projector remains the same (MLP with 2 hidden layers). We compare with two baselines based on TLDR, but which respectively train with a reconstruction (MSE) and a contrastive (Contrastive) loss. Our main baselines are PCA with whitening, and the original 2048-dimentional features (GeM-AP Revaud et al. (2019)), *i.e.* before projection.

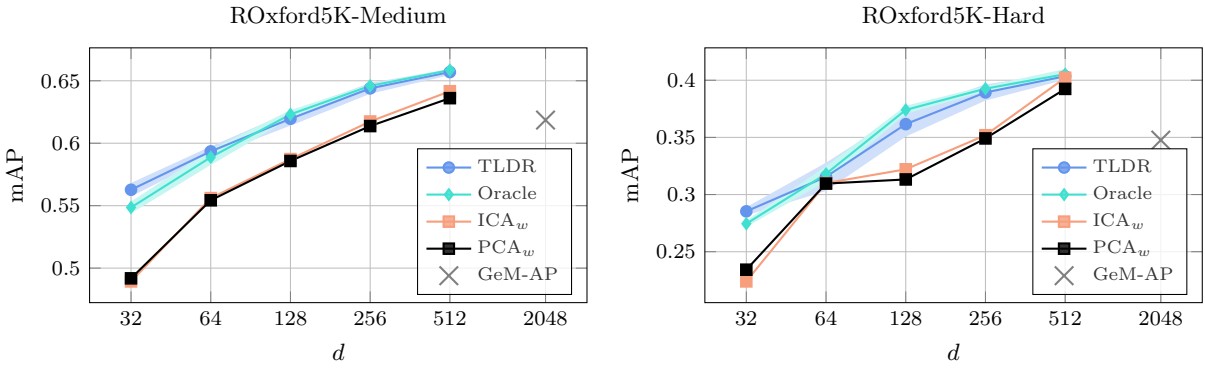

Figure B: **Neighbor-supervised with oracle**. Mean average precision (mAP) on ROxford5K (Radenović et al., 2018a) for the Medium (left) and Hard (right) test sets, as a function of the output dimensions $d$. We compare TLDR with an oracle version that uses labels to select training pairs. We include as baselines both PCA and ICA with whitening, and the original 2048-dimensional features (GeM-AP [32]), i.e. before projection.

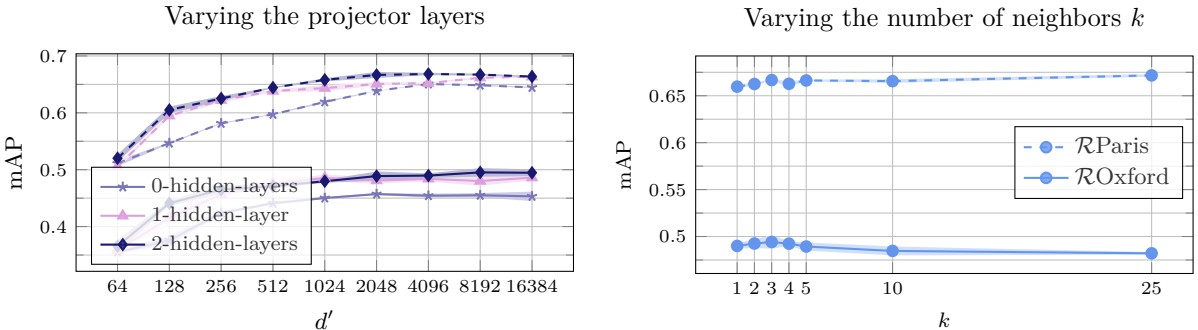

Figure C: **Impact of TLDR hyper-parameters** with a linear encoder and $d = 128$. Dashed (solid) lines are for RParis6K-Mean (ROxford5K-Mean). (Left) Impact of the *auxiliary* dimension $d'$ and the number of hidden layers in the projector. (Right) Impact of the number of neighbors $k$. We see how the algorithm is robust to the number of neighbors used.

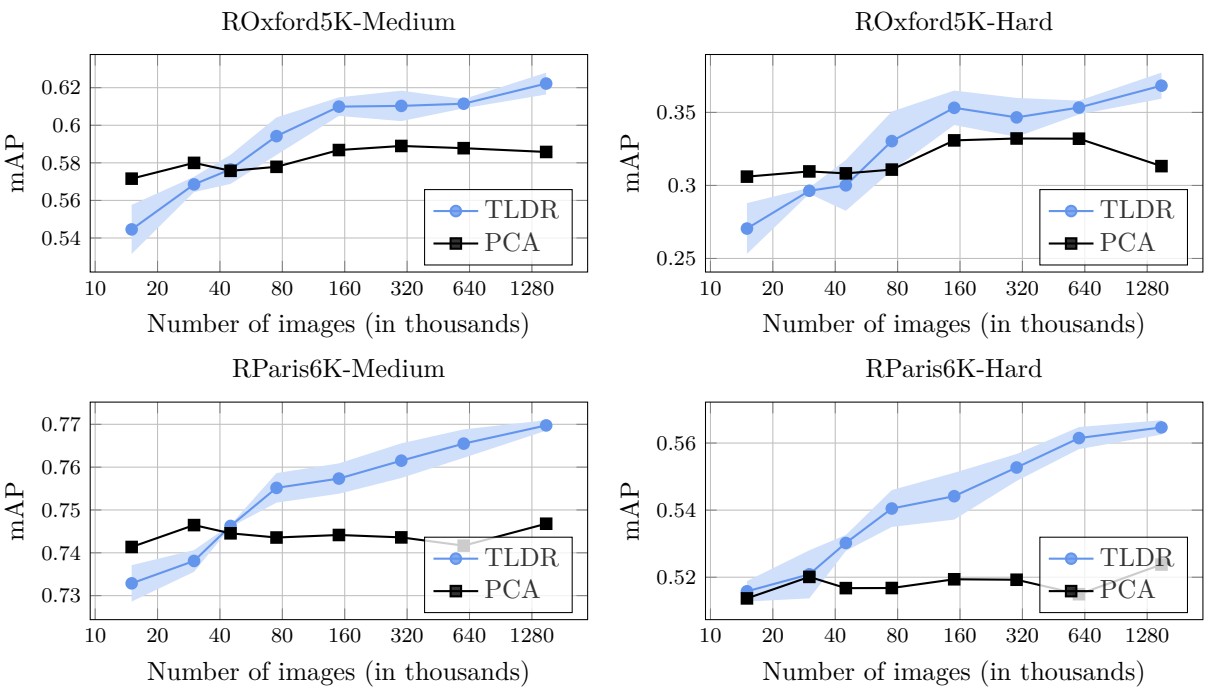

Figure D: **TLDR benefits from larger training sets.** Impact of the size of the training set on performance. TLDR uses a linear encoder and $d = 128$.

## B.4 Varying the size of the training set

In Figure D we show the impact of the size of the training set on TLDR's performance by randomly selecting subsets of images of increasing size from the Google Landmarks training set (Weyand et al., 2020). As we see, PCA outperforms TLDR for a reduced number of images, however, it does not benefit from adding more data, keeping the same performance across all training set sizes. In contrast, TLDR does benefit from adding more data; all plots suggest that a larger training set could potentially boost the performance even further, increasing the gap with respect to PCA.

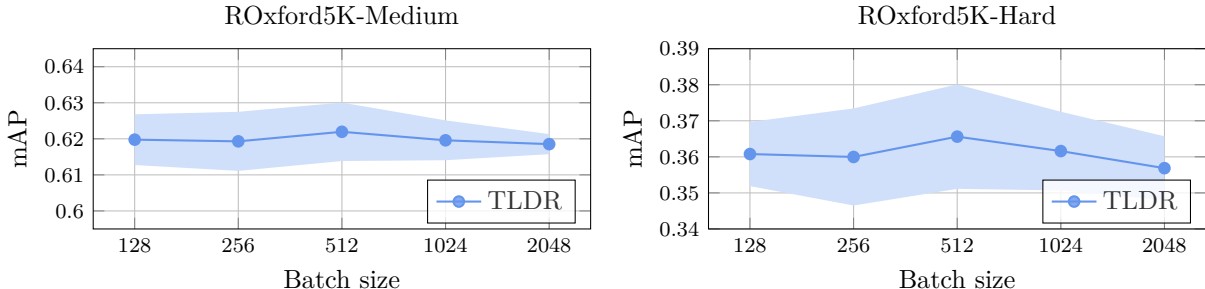

Figure E: **The surprising stability of TLDR across batch sizes**. Impact of the size of the training mini-batch on performance. TLDR uses a linear encoder and $d = 128$.

### B.5    Batch size ablation

Finally, in Figure E, we show results of TLDR varying the size of the training mini-batch. Surprisingly, we observe it is stable across a wide range of values, allowing training TLDR under limited memory resources.

### B.6    Training time comparisons when varyin the size of the training set

In Figure F we report training time when learning a linear dimensionality reduction encoder while varying the size of the training set. We randomly select subsets of images of increasing size from the Google Landmarks training set (Weyand et al., 2020) (1.5M images) and measure the training time and $\mathcal{R}$Oxford performance for TLDR and for a few manifold learning methods. All methods are run on the same servers. We used 16 CPUs and 300GB memory for all manifold learning methods, and one 32GB V100 GPU for TLDR. There is no publicly available easy-to-use GPU implementation for these methods, so we instead use the multi-core versions of scikit-learn.

The tables show that methods like LLE and LTSA scale exponentially with respect to both the time and memory needed. When sampling 5% of the data, *i.e.* 75k images, TLDR (and LPP) require 5 minutes to train while LLE and LTSA require over 4 hours. The LLE and LTSA runs took more than 3 (7) days when subsampling 8% (10%) of the data, while subsampling 15% and above led to out-of-memory (OOM) related crashes despite 300GB of memory available. The LPP method is scaling better, but a) is 5x slower than TLDR when subsampling 10% of the dataset and b) leads to OOM when subsampling 15%. Obviously, reported times highly depend on their implementation. Here, we used the popular scikit-learn multi-core implementations for LLE and LTSA and the best implementation of LPP we could find[8].

### B.7    ResNet-101 features

We also experimented with features obtained from a larger pre-trained ResNet-101 model from Revaud et al. (2019)[9]. We see in Figure G that TLDR retains a significant gain over PCA and in fact surpasses the highest state-of-the-art numbers based on global features as reported in Tolias et al. (2020) for $\mathcal{R}$Oxford.

### B.8    Additional results with DINO features

In Figure Ha we present results on $\mathcal{R}$Oxford using a ResNet-50 DINO pretrained on ImageNet; dimensionality reduction is learned on either ImageNet or GLD-v2. In Figure Hb we report performance on $\mathcal{R}$Paris using a ViT-S/16 DINO pretrained on ImageNet.No labels are used at any stage.

---

[8]https://github.com/jakevdp/lpproj
[9]https://github.com/naver/deep-image-retrieval

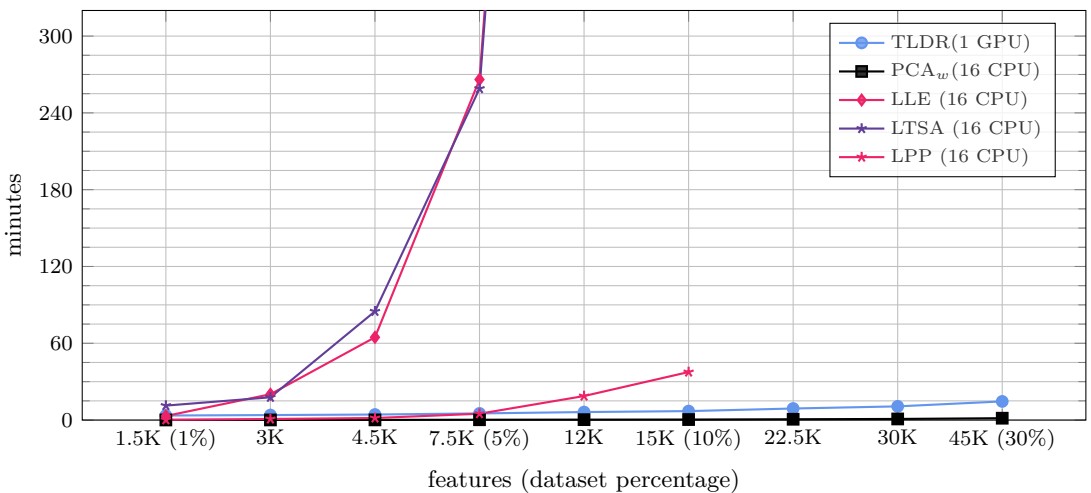

Figure F: **Training time as training dataset size increases** for $d = 32$. TLDR uses a linear encoder, a 2-hidden layer projector with $d' = 2048$ and is trained for 100 epochs.

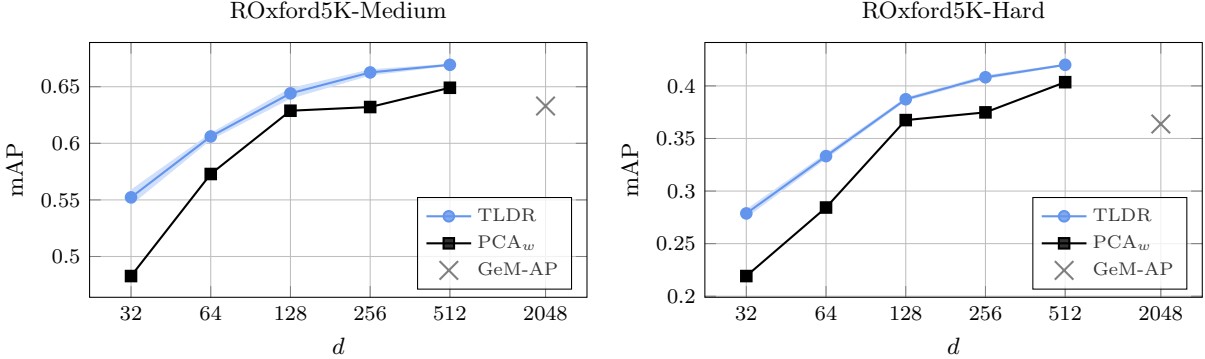

Figure G: **ResNet-101 features**. Mean average precision (mAP) on ROxford5K (Radenović et al., 2018a) for different values of output dimensions $d$, using features obtained from the pre-trained ResNet-101 of Revaud et al. (2019).

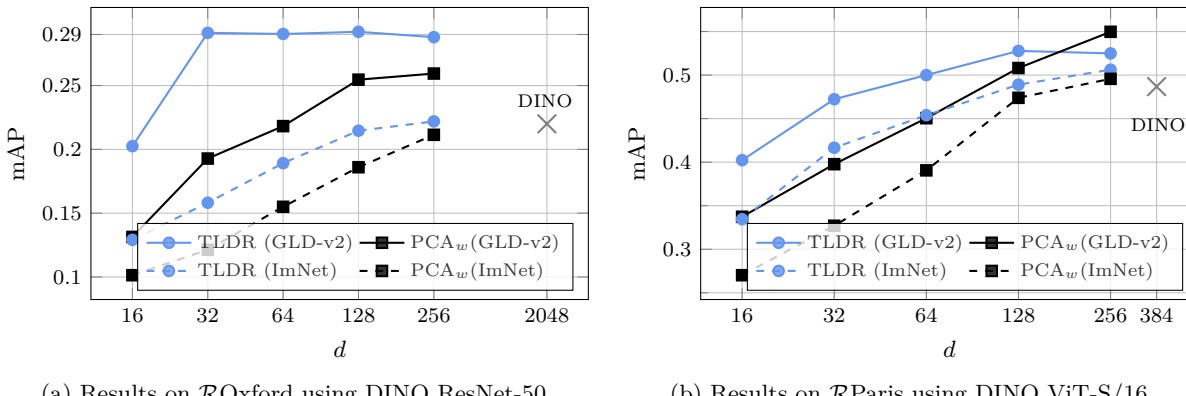

(a) Results on $\mathcal{R}$Oxford using DINO ResNet-50.      (b) Results on $\mathcal{R}$Paris using DINO ViT-S/16.

Figure H: **Self-supervised landmark retrieval performance using a DINO backbones**. Both plots report mAP on $\mathcal{R}$Oxford as a function of the output dimensions $d$. Left: Performance on $\mathcal{R}$Oxford using a ResNet-50 DINO pretrained on ImageNet; dimensionality reduction is learned on either ImageNet or GLD-v2. Right: Performance on $\mathcal{R}$Paris using a ViT-S/16 DINO pretrained on ImageNet. No labels are used at any stage.

Table A: Summary of **tasks and datasets** for the first stage document retrieval experiments presented in the Appendix.

| Dataset | # Documents | # Queries | Avg positives per query | Avg query length | Avg document length | Retrieval type |
|---|---|---|---|---|---|---|
| Question answering (pretraining only) | | | | | | |
| MSMarco | 8.8M | 6980 | 1.1 | 6 | 56 | Asymmetric |
| Argument retrieval | | | | | | |
| ArguANA | 8674 | 1406 | 1 | 193 | 167 | Asymmetric |
| Webis-Touché 2020 | 380k | 49 | 49.2 | 7 | 292 | Asymmetric |
| Duplicate question retrieval | | | | | | |
| Quora | 523k | 5000 | 1.6 | 10 | 11 | Symmetric |
| CQADupStack | 457k | 13145 | 1.4 | 9 | 129 | Symmetric |

# C    Additional experiments: Document retrieval

In Section 3 of the main paper we studied first stage document retrieval under the task of argument retrieval, where both dimensionality reduction and evaluation are performed on datasets designed for the same task. In this section, we extend this evaluation protocol introducing a new task: duplicate query retrieval and now investigate not only dimensionality reduction for the same task, but also the case of *dimensionality reduction transfer*.

In the following paragraphs we first introduce the five datasets we use for first stage document retrieval, and then we discuss the additional experiments involving duplicate query datasets.

## C.1   Tasks and dataset

A summary of dataset statistics is available in Table A and examples for each dataset are available in Table B.

**MSMarco passages (Nguyen et al., 2016):** question and answer dataset based on Bing queries. Very sparse anotation with a high number of false negatives. Queries (Q) and Documents (D) are from different different domains due to size and content. Retrieval is asymmetric, because if you input a D as Q, the answer will not be D. Used only for pretraining as it has a set of training pairs for contrastive learning, while our aim is to perform self-supervision only. For this goal, we have chosen the other four datasets, that do not have

Table B: **Examples of queries and documents** from all the document retrieval datasets we use. Table extracted from Thakur et al. (2021); Note the difference of length between query and document in some datasets.

| Dataset | Query | Relevant-Document |
|---|---|---|
| MSMARCO | what fruit is native to australia | *<Paragraph>* Passiflora herbertiana. A rare passion fruit native to Australia. Fruits are green-skinned, white fleshed, with an unknown edible rating. Some sources list the fruit as edible, sweet and tasty, while others list the fruits as being bitter and inedible. assiflora herbertiana. A rare passion fruit native to Australia... |
| ArguAna | Sexist advertising is subjective so would be too difficult to codify. Effective advertising appeals to the social, cultural, and personal values of consumers. Through the connection of values to products, services and ideas, advertising is able to accomplish its goal of adoption... | *<Title>* media modern culture television gender house would ban sexist advertising *<Paragraph>* Although there is a claim that sexist advertising is to difficult to codify, such codes have and are being developed to guide the advertising industry. These standards speak to advertising which demeans the status of women, objectifies them, and plays upon stereotypes about women which harm women and society in general. Earlier the Council of Europe was mentioned, Denmark, Norway and Australia as specific examples of codes or standards for evaluating sexist advertising which have been developed. |
| Touche-2020 | Should the government allow illegal immigrants to become citizens? | *<Title>* America should support blanket amnesty for illegal immigrants. *<Paragraph>* Undocumented workers do not receive full Social Security benefits because they are not United States citizens " nor should they be until they seek citizenship legally. Illegal immigrants are legally obligated to pay taxes... |
| CQADupStack | Command to display first few and last few lines of a file | *<Title>* Combing head and tail in a single call via pipe *<Paragraph>* On a regular basis, I am piping the output of some program to either 'head' or 'tail'. Now, suppose that I want to see the first AND last 10 lines of piped output, such that I could do something like ./lotsofoutput | headtail... |
| Quora | How long does it take to methamphetamine out of your blood? | *<Paragraph>* How long does it take the body to get rid of methamphetamine? |

a readily available set of training pairs (or triplets) for training, and thus self-supervision or unsupervised learning is required.

**ArguAna (Wachsmuth et al., 2018):** Counter-argument retrieval dataset. Queries and documents belong to the same domain, with some queries being a part of the corpus, which makes it not suitable for training on this dataset. Queries and documents come from the same domain in both size and content, however associated query-document pairs have inverse context (Q defends a point, D is a rebuttal of Q), so input Q should not retrieve Q, if it is on the database. Retrieval is asymmetric as a query should not retrieve itself.

**Webis-Touché 2020 (Bondarenko et al., 2020; Wachsmuth et al., 2017):** Argument retrieval dataset. Queries and documents are from different domains due to size and content, with queries being questions and documents being support arguments for the question. Retrieval is asymmetric as a query should not retrieve itself.

**CQADupStack (Hoogeveen et al., 2016):** Duplicate question retrieval from StackExchange subforums, composed of 12 different subforums. Corpuses are concatenated during training and mean result over all corpuses is used for testing (*i.e.* every corpus has equal weight even if the number of queries is different). Queries are titles of recent submissions, while documents are concatenation of titles and descriptions of existing ones. Queries and documents are from different domains due to size and content, with the query domain being a part of the document one (Queries are contained in the documents). Retrieval is symmetric as a query should return itself.

**Quora:** Duplicate question retrieval from the Quora platform. Queries are titles of recent submissions, while documents are titles of existing ones. Queries and documents are from the same domain concerning size and content. Retrieval is symmetric as a query should return itself.

## C.2 Experimental results

We now test different combinations of the previously introduced datasets as dimensionality reduction and test datasets. The setup for all experiments is the same as the experiments in the main paper. In order to summarize our results (9 combinations of train/test datasets), we consider $d = 64$ (second highest we investigate) as the comparison mark between PCA and TLDR. If TLDR outperforms PCA for all $d \leq 64$, we consider that it performed better than PCA, and otherwise we consider that PCA performed better than TLDR. Note that in all cases the lower the dimension the better TLDR performed against PCA. We also

Table C: **Summary of the results on document retrieval.** (L=0) and (L>0) indicate which version of TLDR had better perfomance (linear and factorized linear respectively). Note that arguana is not suitable for training (not represented) and that we are not interested in using the same dataset for dimensionality reduction and test (thus the empty cells).

| | | | Test dataset | | | |
| | | | Argument retrieval | | Duplicate query | |
| | | | ArguAna | Webis-Touché 2020 | Quora | CQADupStack |
| Dimensionality reduction | Argument Retrieval | Webis-Touché 2020 | TLDR (L>0) | | PCA (L>0) | PCA (L=0) |
| | Duplicate Question | Quora | TLDR (L>0) | TLDR (L=0) | | PCA (L=0) |
| | | CQADupStack | TLDR (L>0) | TLDR (L=0) | PCA (L>0) | |

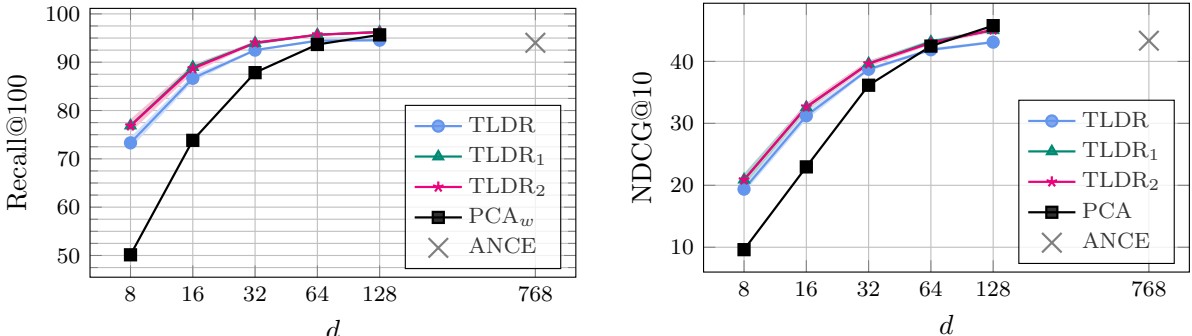

Figure I: **Argument retrieval results on the ArguAna dataset using Webis-Touché 2020 for dimensionality reduction** for different values of output dimensions $d$. On the left we present Recall@100 and on the right we present NDCG@10.

report which version of TLDR performed better, $L > 0$ means that factorized linear is better than linear and $L = 0$ the opposite. We present a summary of the experimental results in Table C, and provide depictions of some experiments in Figures I through L. From the results presented on the table, we derive two conclusions about TLDR:

1. **Differences in retrieval from pretraining to dimensionality reduction impacts results :** Looking into evaluation on argument retrieval, TLDR outperforms PCA. On the other hand, looking into evaluations on duplicate query retrieval, PCA is always able to outperform TLDR for $d \geq 64$. We infer that this must be derived from the difference in retrieval condition, as in all tests with asymmetric retrieval TLDR is able to outperform PCA. Note that symmetric retrieval and same domain for document and queries differs from the original pretraining task, and we posit that PCA is more robust to this type of change (which does not happen in our image retrieval experiments). Although we have this initial suspicion validated with 4 datasets a proper conclusion would need more dataset-pairs for experimentation, which we leave for future work.

2. **Choosing linear or factorized linear depends on the statistics of the dataset:** Analyzing the results we are able to detect that the choice of which version of TLDR one should use depends on the length of queries and documents of the original dataset. If both lengths are equal, factorized linear is better (ArguANA and Quora), if not then linear is the better choice (Webis-Touché 2020 and CQADupStack). Even if by using ANCE representations we should not need to deal with these differences (we only tackle embeddings of fixed size), the statistics of the resulting embedding is different enough that it is detected by the batch normalization layer that is added for factorized linear.

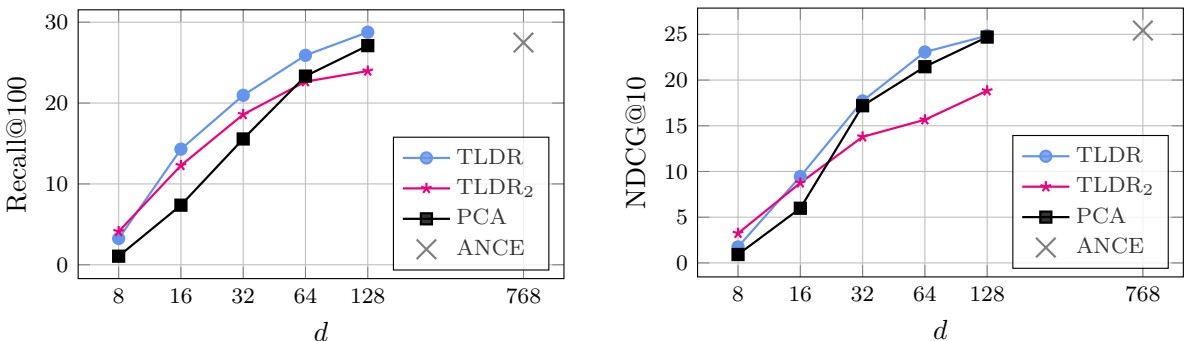

Figure J: **Argument retrieval results on the Webis-Touché 2020 dataset using Quora for dimensionality reduction** for different values of output dimensions $d$. On the left we present Recall@100 and on the right we present NDCG@10.

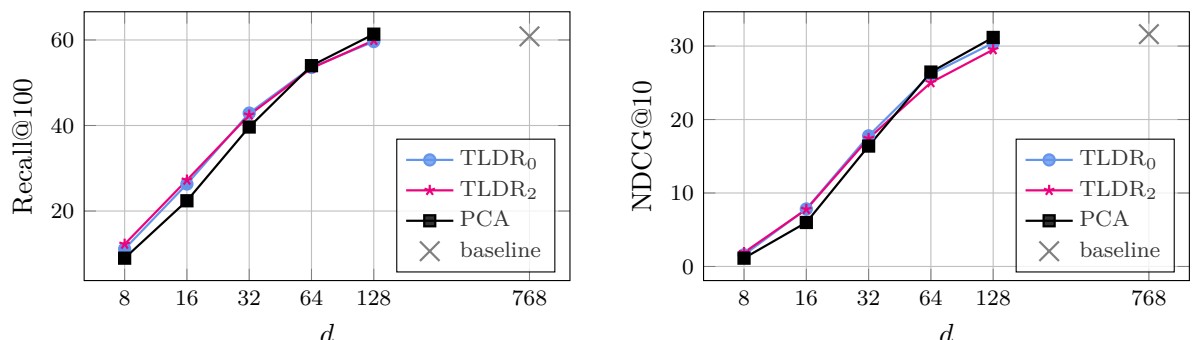

Figure K: **Duplicate question retrieval results on the CQADupstack dataset using Quora for dimensionality reduction** for different values of output dimensions $d$. On the left we present Recall@100 and on the right we present NDCG@10.

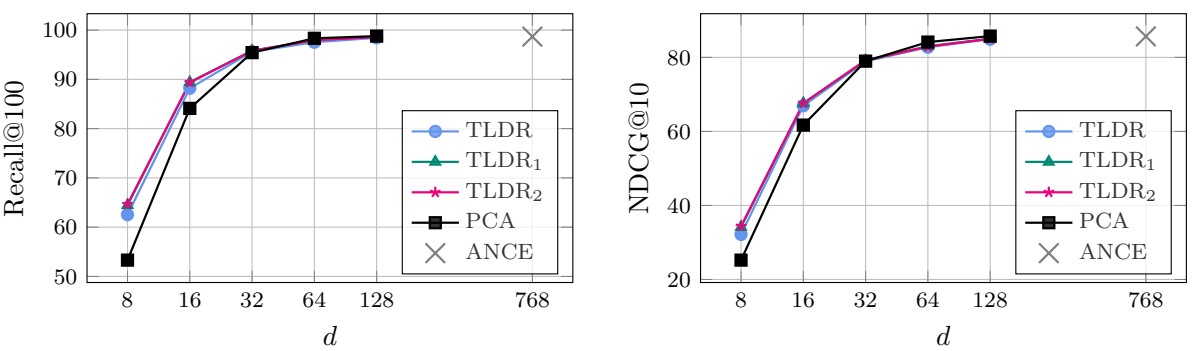

Figure L: **Duplicate question retrieval results on the Quora dataset using Webis-Touché 2020 for dimensionality reduction** for different values of output dimensions $d$. On the left we present Recall@100 and on the right we present NDCG@10.

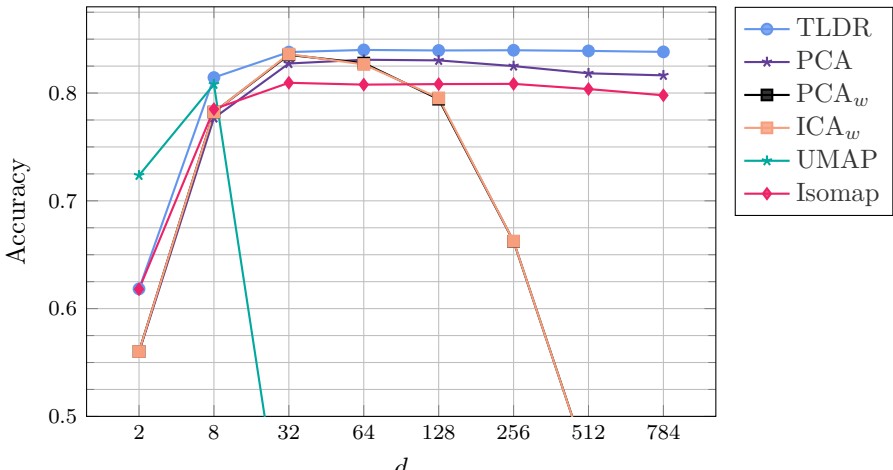

Figure M: **Results on the FashionMNIST dataset** as a function of the output dimensions $d$. We compare TLDR with PCA, PCA with whitening, UMAP and Isomap and report accuracy after $k'$-NN classifiers (with $k' = 100$) following (McInnes et al., 2018). For TLDR and UMAP we set the number of neighbors $k = 100$. The performance of UMAP was very low for $d' > 32$.

## D   FashionMNIST: Learning from raw pixel data and visualization

Although beyond the scope of what TLDR is designed for (see also discussion at the end of Section 4), in this section we present some basic results when using it for learning from raw pixel data and for visualizations, *i.e.* when reducing the output dimension to only $d' = 2$.

**Learning from raw pixel data.** In Figure M we present results when learning directly from raw pixel data. We use the predefined splits and, following related work (McInnes et al., 2018), we measure and report accuracy after $k$-NN classifiers. We see that TLDR retains its gains over any other manifold learning method we tested. We have to note however that these results have to be taken with a pinch of salt, as a) the input pixel space is relatively simple compared to higher resolution natural images and b) to achieve such results we use the prior knowledge that we only have 10 classes and set high values for hyper-parameter $k$, *i.e.* $k = 100$ for all methods compared.

**2D visualizations.** Let us first clarify that TLDR was not created with 2D outputs in mind; in fact, there are other excellent choices for visualization like $t$-SNE, UMAP (McInnes et al., 2018), TriMAP (Amid & Warmuth, 2019) or the recent Minimum-Distortion Embedding (MDE) (Agrawal et al., 2021) that we would use instead. In Figure N we show 2d visualizations when reducing the 60k training set of FashionMNIST to $d = 2$ dimensions. We present results for TLDR, $t$-SNE (Van der Maaten & Hinton, 2008), UMAP (McInnes et al., 2018) and PyMDE (Agrawal et al., 2021). It is interesting how TLDR seems to be optimized for linear separability even for 2-dimensional outputs. For visualizations, we used the pyMDE library[10] provided by the authors of (Agrawal et al., 2021).

## E   Further discussions and related works

**Graph diffusion for query expansion.** For the task of retrieval, assuming access to the search (test) database, methods like (Iscen et al., 2017; 2018a; Liu et al., 2019) utilize manifold learning on the the $k$-NN graph of the database to facilitate *query expansion*. We note that while these methods have shown great empirical performance on the same image retrieval datasets as we experiment on, we do not directly compare to them as their methodology and goals greatly differs from ours. We aim at being *invariant to the target dataset* (thus not performing learning on them), differently from the aforementioned methods they need

---

[10]https://pymde.org/

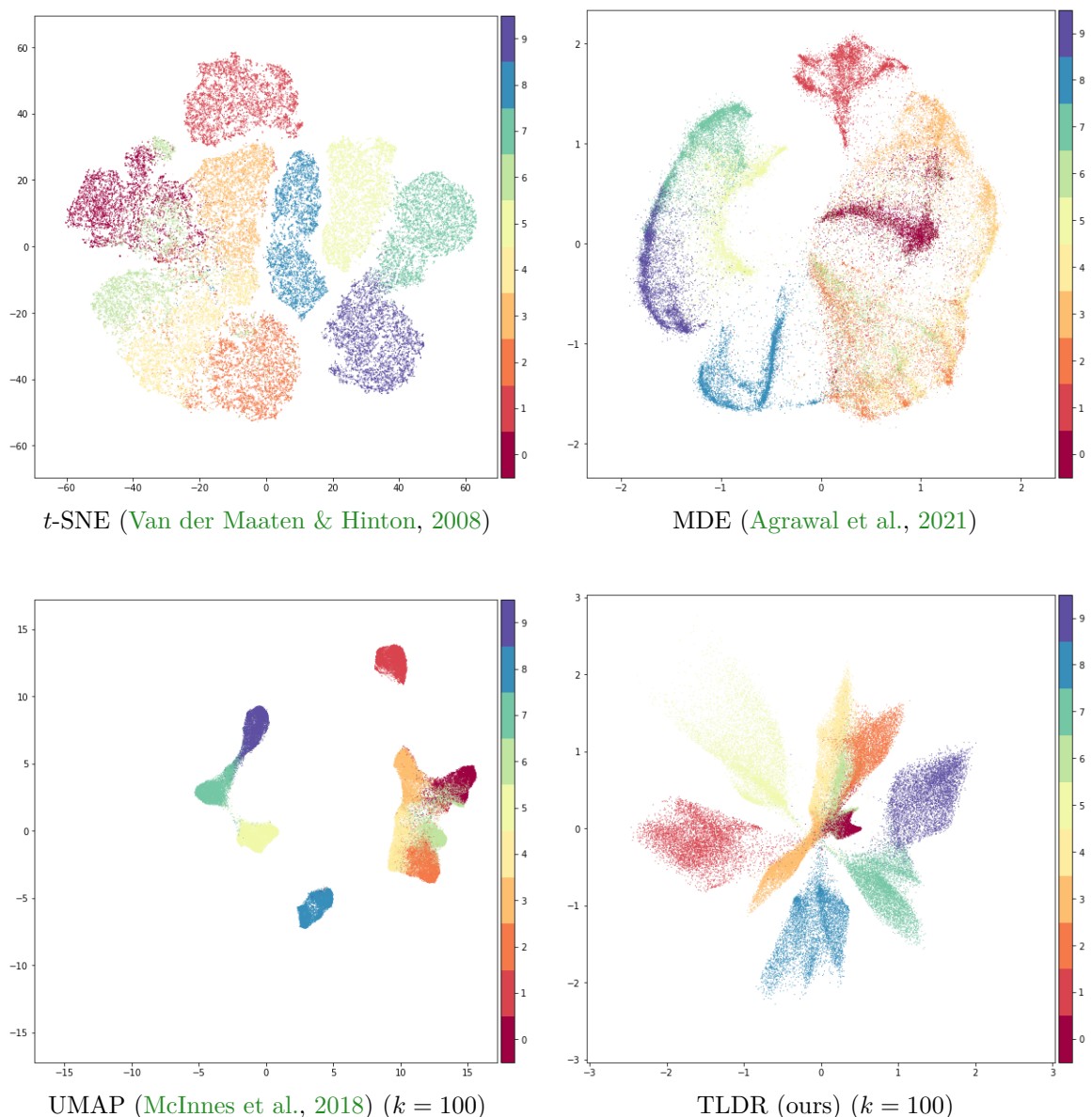

*t*-SNE (Van der Maaten & Hinton, 2008)

MDE (Agrawal et al., 2021)

UMAP (McInnes et al., 2018) ($k = 100$)

TLDR (ours) ($k = 100$)

Figure N: **2D visualizations of the training set of FashionMNIST**. From top to bottom and left to right: *t*-SNE, MDE, UMAP and TLDR.

access to the target dataset for learning, and to its *k*-NN graph during testing. TLDR is complementary to such graph diffusion techniques for query expansion.

**Relation to knowledge distillation.** Knowledge distillation (KD) (Hinton et al., 2015) aims at transferring knowledge from a pre-trained teacher network to a student one, often for neural network compression. One way to perform KD is relational KD (RKD) (Park et al., 2019; Tian et al., 2020; Lin et al., 2020), which transfers knowledge using relations between samples such as distance and angles. TLDR can be seen as a method for RKD. It enforces the student network (encoder) to reproduce a relational property (neighborhood) found on the teacher (the input space). However, there are some main differences to traditional distillation methods: i) the application: self-supervised retrieval instead of supervised classification (Hinton et al., 2015), contrastive (Tian et al., 2020; Lin et al., 2020) or self-supervision for classification (Fang et al., 2021), ii) the definition of the relations: abstract (neighbors), instead of measurable ones (distance, angle), which avoids normalization problems due to the dimensionality difference between teacher and student, and iii) the link

---

**Algorithm 2:** PyTorch-style pseudocode of TLDR.

```
# X: training set of M D-dimensional vectors
# D: input dimension
# d': projector's output dimension
# d: encoder's output dimension

# B: batch size
# k: number of neighbours
# compute_knn_graph: calculates the k nearest neighbours of each vector
# RandomBatchSampler: randomly samples batch_size indices

# Initialization
model = initialize_model()  # initialize both encoder f and projector g
N = compute_knn_graph(X, k)  # returns matrix of size Mxk

# Training
for e in num_epochs:
  for indices in RandomBatchSampler(len(X), batch_size=B):
    x = X[indices]  # BxD
    y = X[random_sample(N[indices], 1)]  # BxD
    x = model(x)  # Bxd'
    y = model(y)  # Bxd'
    loss = BarlowTwinsLoss(x, y)
    loss.backward()
    model.update()

model.discard_projector()  # discard projector g
Z = model(X)  # Mxd
```

---

between teacher and student: in our case, the student becomes a part of the teacher network at the end, instead of being a separate network.

**Relation to node embedding.** Node embedding methods aim at generating representations to graph nodes that are representative of the sample and its relations on the graph. In that sense, TLDR could be seen as learning embeddings for nodes on a graph. Compared to the traditional methods in this space, such as LINE (Tang et al., 2015), Node2Vec (Grover & Leskovec, 2016), DeepWalk (Perozzi et al., 2014), TLDR has three clear differences: i) does not rely on the edge strength; ii) regularization of the space based on the decorrelation of dimensions instead of L2-norm or orthogonality; and iii) only the 1-hop neighborhood information is used. More recent node embedding solutions are based on deep learning architectures that incorporate diffusion properties in the architecture like GCNs (Kipf & Welling, 2017; You et al., 2020), while TLDR achieves a similar effect via the Barlow Twins loss.

### E.1   Limitations of our work

In the context of document retrieval we also tested TLDR on another task: duplicate question retrieval. In duplicate question retrieval TLDR was only able to outperform PCA for the lower dimension values (d=8,16,32). We posit that TLDR does not achieve significant gains for the rest of the dimensions because duplicate task differs too much from the original pretraining task (QA on MSMarco dataset) in that the duplicate retrieval is symmetric (the documents retrieved by a query should also appear when we use the document as query), while pretraining and argument retrieval is assymetric. In order to verify this, we performed ablations with different pairs of (dimensionality reduction,target dataset) and confirm that if the target dataset is a duplicate retrieval task TLDR is not able to outperform the compared method, but if we use duplicate retrieval only for dimensionality reduction and test on argument retrieval TLDR is able to outperform the compared methods. For full discussion and results *cf.* Section C.

## F   Pseudocode of TLDR

In Algorithm 2 we show the pseudocode of TLDR, which includes initialization, training, and projection.

