# OpenReview forum: "TLDR: Twin Learning for Dimensionality Reduction"
_TMLR — Accepted by TMLR_

### Review · Reviewer_FuBN · 2022-04-14

**Summary Of Contributions:**

This paper proposes an unsupervised dimensionality reduction method called TLDR. The method first defines a distance function in the input space. Then k-NN is used to find k nearest inputs for any sample x. These k samples serve as the positive pairs for x. For each pair (x, y), both of them are first mapped to a low-dimensional latent space and then project back to a high-dimensional space. In this high-dimensional space, the authors compute a Barlow Twins loss as the objective function. Intuitively, the loss function forces the projected features to be highly correlated. This process requires no supervision. Once the training finishes, the encoder can be used for dimensionality reduction. Authors apply the method to image and document retrieval tasks. The results are better than existing methods.

**Broader Impact Concerns:**

I don't have concerns on the ethical implications of this work.

**Requested Changes:**

1. Better explanation/connections with existing works like Barlow Twins, CCA, BYOL, etc.
2. How the encoder is used should be more clear. It's better to use some math notations to explain.
3. It looks like you only attempted euclidean distance, could you also try some other distances?
4. Is the euclidean distance complementary to the common augmentation techniques like distortion or shifting?

**Strengths And Weaknesses:**

Pros:
1. The paper is well-organized and the method itself is not hard to understand.
2. Exhaustive experiments have been performed to validate the performance. The results look sound.
3. Ablation studies on projector architecture, number of neighbors, and batch size are comprehensive.

Cons:
1. Authors should further explain why they use euclidean distance. Let's say we shift an image to the right and bottom a lot. It is essentially the same image, but the euclidean distance would increase a lot. Though euclidean distance can be changed to other distances, if we have to choose different distances for different tasks, then the distance selection would become another hyper-parameter.
2. Essentially the method is still looking for a certain invariance. The key is the distance defined in the input space. TLDR can be viewed as a variant of BYOL where the loss function is changed to Barlow Twins loss and the augmentation is replaced with k-NN. Further comparison and discussion with BYOL should be presented. Also, compared with the original Barlow Twins paper, this paper's novelty is incremental.
3. The exact tasks should be better explained. Do you first pretrain the TLDR model without supervision and then perform supervised finetuning? How do you evaluate the information retrieval performance? Compared to the general supervised tasks, is the o
4. What is the setup for contrastive loss and how do you make sure the comparison is fair? Should you also include BYOL for comparison?
5. If you have to compute the correlation matrix for each pair (x, y), wouldn't the time cost be huge?
6. Barlow Twins seems to be very closely related to CCA. Some related papers are worth discussing, DCCA[1], DCA[2], DAPC[3].

[1] Andrew, Galen, et al. "Deep canonical correlation analysis." International conference on machine learning. PMLR, 2013.
[2] Clark, David, et al. "Unsupervised discovery of temporal structure in noisy data with dynamical components analysis." Advances in Neural Information Processing Systems 32 (2019).
[3] Bai, Junwen, et al. "Representation Learning for Sequence Data with Deep Autoencoding Predictive Components." International Conference on Learning Representations. 2020.

---

> ### Author Response · Authors · 2022-04-20
> **Response to Reviewer FuBN (1/3)**
>
> We thank the reviewer for their constructive review and positive comments. Below we first provide what we think is an important clarification on the goal of our method, and then answer all questions. We welcome any further discussion. Note that we intend to upload an updated version of the manuscript after getting feedback from and discussing with all reviewers.
>
> __Overall goal: dimensionality reduction vs representation learning__
>
> First and foremost, we would like to clarify the goal of the method proposed in our manuscript, and emphasize the fact that it does not tackle (self-supervised) representation learning, but **dimensionality reduction**.
>
> Methods for self-supervised representation learning (SSL) like BYOL, DINO or Barlow Twins:
> * start from structured inputs of a specific nature (i.e. a 2D grid of pixel intensities)
> * learn a large encoder with hundreds of millions of parameters that transforms this input into into a vector representation useful for several downstream tasks
>
> Our method focuses on the task of dimensionality reduction. As such it
> * starts from any trustworthy, potentially blackbox, representation
> * learns a simple (linear or MLP) encoder and without any further assumptions on the nature of the input space beyond the fact that structure should be preserved.
>
> Dimensionality reduction is a very common task, used in many practical applications ranging from biology to AI. This is because, more often than not, learning a small encoder on top of an existing representation is easier than training a lower dimensionality representation end-to-end:
> * End-to-end training requires large resources, especially for the recent foundation models composed of millions or billions of parameters
> * Fine-tuning those models to certain domains or tasks require non trivial knowhow
>
> In several cases, training/fine-tuning is simply not possible as features are produced by a blackbox. Consequently, dimensionality reduction methods like PCA are still part of state-of-the-art pipelines e.g. for image retrieval (see Weinzaepfel et al ICLR  2022).
>
> In our manuscript, we propose TLDR as a post-hoc dimensionality reduction method, on top of robust representations. TLDR leverages losses introduced for visual SSL, but for a far more generic task. We show that TLDR provides practitioners with an easy way of not just compressing, but also improving the performance of large state-of-the-art models without the need to finetune large representation learning encoders. It is much simpler to treat a large model as a feature extractor, save the features and then only learn a simple (linear or MLP) encoder to get compressed vectors more suited to the desired tasks or dataset. As we show, one can get significant gains and compression when starting from e.g. DINO features.
>
> This makes an obvious **link between representation learning and dimensionality reduction**: we could use the _output_ of models like Barlow Twins, BYOL, DINO, BERT or any representation learning method as the _input_ for our TLDR encoder. Yet, those two steps are not interchangeable and we focus on the latter.
>
> We agree with the reviewer that the motivation above could be made clearer, _we will make sure to clarify and update the text accordingly_.

---

> > ### Author Response · Authors · 2022-04-20
> > **Response to Reviewer FuBN (2/3)**
> >
> > __Responses to reviewer questions__
> >
> > > Authors should further explain why they use euclidean distance. Let's say we shift an image to the right and bottom a lot. It is essentially the same image, but the euclidean distance would increase a lot. [...] Is the euclidean distance complementary to the common augmentation techniques like distortion or shifting?
> >
> > When operating on high-dimensional vector spaces, it is a standard assumption shared by all manifold learning methods and therefore also by TLDR that there exists a locally linear/euclidean manifold in the input space that we want to preserve.
> >
> > As discussed above, augmentations like the ones used in visual representation learning are based on the fact that we know that the source signal is pixels and therefore e.g. visual semantics should be preserved after shifting, spatial cropping, or color jittering. In our case, we _start from vectors in high-dimensional spaces_ without further assumptions, i.e. we assume that the input space is robust and invariant to common perturbations like the ones mentioned above for images. For TLDR it doesn't matter if the vectors represent images or documents; we do not adapt our dimensionality reduction method for input spaces of a different nature (i.e. visual or textual). As we discuss on Page 12 ("TLDR out of its comfort zone") if one is able to define stronger priors for representation learning, it makes sense to use them; TLDR is not a replacement for those and could be complementary (e.g. for visual representation learning “augmentation invariance is a much more suited prior in that regard”).
> >
> >
> > > the distance selection would become another hyper-parameter. [...]
> >
> > It is true that TLDR is agnostic to the way neighbors are extracted, and other similarity measures could be used, for instance to take into account prior information. TLDR will then learn an embedding (output) space for which the Euclidean distance is to be used, something really desirable because of its simplicity. _We thank the reviewer and will add a short discussion in Section 4._
> >
> > >  TLDR can be viewed as a variant of BYOL where the loss function is changed to Barlow Twins loss and the augmentation is replaced with k-NN. Further comparison and discussion with BYOL should be presented. [...] Should you also include BYOL for comparison?
> >
> > The answer to this is related to the earlier discussion about the goal of our method. TLDR is not to be seen as a replacement for Barlow Twins (BT) or BYOL, but instead a method that can be used on top of such representations to reduce their dimensionality.
> >
> > It is true that we could devise a variant of TLDR that uses the BYOL loss instead of the BT one. We chose the latter because: a) according to comparisons in the BT paper, the two perform similarly, with BT achieving slightly better performance for SSL tasks; b) BT has similar properties to BYOL, both are non-contrastive and in fact the first term of the BT loss is very close to what BYOL is optimizing; c) the second term not only gives BT an elegant way of avoiding collapse, but de-correlating the output space is a process that highly suits the dimensionality reduction task; d) from our experiments and ablations we saw that the BT loss was highly robust to most hyperparameters out-of-the-box.  _We will add a discussion about BYOL in Section 4._
> >
> > >  Also, compared with the original Barlow Twins paper, this paper's novelty is incremental.
> >
> > We agree that the _technical_ novelty is “incremental”. Our design choices, however, enable TLDR to be used in a far more generic setting than the one presented in the Barlow Twins paper, and e.g. use a powerful loss like Barlow Twins in cases beyond visual representation learning and for compressing document vectors out of the box. We see TLDR as a generic and powerful tool that can be useful for applications beyond visual representation learning, wherever dimensionality reduction is used. By leveraging recent advances from SSL, TLDR offers a modern, highly scalable and better performing alternative to the dimensionality reduction methods used in practice.

---

> > > ### Author Response · Authors · 2022-04-20
> > > **Response to Reviewer FuBN (3/3)**
> > >
> > > > The exact tasks should be better explained. Do you first pretrain the TLDR model without supervision and then perform supervised finetuning? How do you evaluate the information retrieval performance? [...] How the encoder is used should be more clear. It's better to use some math notations to explain.
> > >
> > > This is a very good point. _We will add details on the evaluation process in the experiments section._
> > >
> > > Let us first note that in all cases we start from feature vectors; It is a pre-requirement for the dimensionality reduction task to encode any data not in vector form (images/documents) with some representation learning model into a $D$ dimensional input space where, without loss of generality, the Euclidean distance is meaningful at least locally. For the experiments presented in the manuscript, we use the DINO and AP-GeM models as feature extractors to encode images for visual tasks, and the ANCE BERT model for documents.
> > >
> > > We never use any supervision for learning TLDR or any of the methods we compare to. Given a learned dimensionality reduction encoder, we then encode all downstream task data for each test dataset, and evaluate them in a “zero-shot” manner, using non-parametric classifiers (k-NN) for all retrieval tasks. Specifically, for landmark image retrieval we use the common protocols from ROxford/RParis, where specific queries are defined; for every query, we measure the mean average precision metric across all other images depicting the same landmark. For the ImageNet case, we follow the exact process used in DINO and other papers: Each image in the val set is used as a query for each class, and the “database” consists of the full validation set across all 1000 classes. We once again use k-NN to compute a ranked list for the database, and aggregate the labels from the top-20 ranked database images for each query, assigning the most prominent class as the predicted one.
> > >
> > > > If you have to compute the correlation matrix for each pair (x, y), wouldn't the time cost be huge?
> > >
> > > There is indeed some added cost, e.g. compared to using a loss like BYOL, that doesn’t require computing the correlation matrix. This is in practice an O(d’^2) cost per pair, a cost that is non-negligible, but also not a problem when you only learn linear or MLP encoders.
> > >
> > > In term of computation, it is worth noting that the learning a dimensionality reduction encoder for TLDR is generally orders of magnitude less expensive than than e.g. visual representation learning: We always start from pre-extracted feature vectors during training and only forward and backpropagate through a linear or MLP encoder and an MLP projector. There is no transformer or CNN backbone to learn or even keep in GPU memory and even datasets like GLD-v2 (composed of 1.5M vectors in 2048 dimensions for the AP-GeM model) can easily fit in system memory. _We will clarify this in the paper._
> > >
> > >
> > > > Barlow Twins seems to be very closely related to CCA. Some related papers are worth discussing, DCCA[1], DCA[2], DAPC[3].
> > >
> > > This is a great point. The MLP variant of TLDR with the BT loss is definitely close to DCCA and BT can be seen as a more modern, simpler variant of the latter: the addition of the projector seems to be making a large difference, while deriving the gradient of DCCA involves for example the gradient of the trace of the matrix square-root, something that is not needed in the BT loss. Moreover, DCCA showed their best results using full-batch optimization and L-BFGS, a far less scalable setting. We thank the reviewer for bringing DAPC and DCA to our attention; these are indeed related methods tailored for time-series data that we were unaware of and use analogous ideas around “temporal” neighbors.
> > > We thank the reviewer and _will add a paragraph in Section 4 discussing relations to CCA and the “deep” variants the reviewer mentions above_.

---

### Review · Reviewer_j6W2 · 2022-04-20

**Summary Of Contributions:**

The authors propose a dimensionality reduction technique (twin learning for dimensionality reduction, or TLDR). The quality of the new representations is mainly measured by performance in retrieval tasks. The main bulk of experiments demonstrate that TLDR yields better representations than principal component analysis (PCA).

**Requested Changes:**

The following requested changes are necessary for securing my recommendation for acceptance.

* Compare the runtime of TLDR with that of PCA.
* Explain how PCA is done on the dimensionality reduction training set (since the dataset sizes are really big).
* Explain how the error bars (like those in Figure 2) are generated.
* Explain why some methods have error bars while some do not (in Figure D, and other figures, PCA does not have error bars while TLDR does).


**Strengths And Weaknesses:**

# Strengths
The evidence supporting the claim that TLDR representations are more useful than PCA representations is strong. Figures in the main text cover a diverse set of tasks. It is consistent across figures that a flavor of TLDR outperforms PCA. Figures in the appendix show that TLDR is insensitive to its own algorithmic hyperparameters, making the method easy to use.

The paper also cites relevant benchmarks and adequately discusses whether alternative approaches could/should be applied to the tasks at hand (Section 4).

# Weaknesses
There are some minor weaknesses which should not take long to address.

The foremost weakness is as follows. The paper makes the claim that TLDR is a scalable method (Section 1 says “This leads to a highly scalable method”). The evidence for this claim is not strong. For instance, the paper does not include discussions of runtime, especially in relation to the runtime of the main baseline, PCA. It is expected that TLDR takes more time than PCA; it would be good to know how much longer.

The other noticeable weakness is that the paper does not explain clearly how exactly PCA is done on the dimensionality reduction training set. Based on the xticks of Figure D from the appendix, sometimes the dimensionality reduction training set size can be as large as 1 million. As far as I know, standard singular value decomposition techniques for PCA cannot process such a large dataset in one pass.

---

> ### Author Response · Authors · 2022-04-22
> **Response to Reviewer j6W2**
>
> We thank the reviewer for their positive and constructive review. Below we provide some clarifications and respond to the reviewer’s questions. We welcome any further discussion. Note that we intend to upload an updated version of the manuscript after getting feedback from and discussing with all reviewers.
>
> __Detailed Responses__
>
> > The paper makes the claim that TLDR is a scalable method (Section 1 says “This leads to a highly scalable method”). The evidence for this claim is not strong. For instance, the paper does not include discussions of runtime, especially in relation to the runtime of the main baseline, PCA. It is expected that TLDR takes more time than PCA; it would be good to know how much longer. [...] Compare the runtime of TLDR with that of PCA.
>
> By “scalability” we refer to the ability of the proposed TLDR to be efficiently applied to arbitrary large datasets and output spaces. We believe this to be an important property of TLDR, given that most manifold learning methods are unable to "scale" to large datasets or output dimensions: many require constructing and navigating large graphs, computing eigendecompositions of large laplacians or other optimization processes that scale super-linearly with either the output dimension or dataset size. This is why for example in Fig 6a we had to subsample the datasets to make some of the competing methods fit in memory and/or converge within a couple of days. TLDR uses mini-batch Stochastic Gradient Descent and has linear time and memory complexity with respect to both the training database size and the output dimension $d$. TLDR is a “highly scalable” manifold learning method in that regard. We agree that there exist approximate variants of PCA that are also highly scalable (see response below), and this is one if the main reasons PCA is widely used for large-scale retrieval. PCA is however restricted to learning a linear projection, while TLDR can further be used to learn non-linear encoders out-of-the-box.
>
> When it comes to training time, TLDR is noticeably slower than PCA. For example, for $d=128$ and for the GLD-v2 dataset (1.5M vectors), learning PCA takes approximately 18 minutes (multi-core), while 100 epochs of learning linear TLDR with a modest projector (that outperforms PCA) takes approximately 63 minutes (GPU). The latter includes computing (exact) k-NNs for the dataset, which takes approx 13 minutes.
>
> Although slower at training time, linear TLDR __highly outperforms PCA__, while all other manifold learning methods that we compare to can neither reach the performance of PCA nor be applied to large training datasets for $d > 32$. More importantly, although the training time of TLDR is higher than PCA,  both methods share the same _linear encoding complexity and time_ during testing, something highly important as this is a process that is repeated on-line at every single retrieval request.
>
> _We thank the reviewer; we will clarify what we mean by scalability and we will add training time comparisons in the updated version._
>
>
> > The other noticeable weakness is that the paper does not explain clearly how exactly PCA is done on the dimensionality reduction training set. [...] standard singular value decomposition techniques for PCA cannot process such a large dataset in one pass. [...] Explain how PCA is done on the dimensionality reduction training set (since the dataset sizes are really big).
>
> We were able to run PCA on millions of data points and hundreds of dimensions using out-of-the-box tools; more precisely, we use the PCA implementation from scikit-learn [A]. For large matrices (> 500x500, which is our case) it uses the randomized SVD approach by Halko _et al_ [B], which is an approximation of the full SVD, but it has been the standard way of scaling PCA to large matrices. _We will clarify this in the text_.
>
> [A] https://scikit-learn.org/stable/modules/generated/sklearn.decomposition.PCA.html
>
> [B] Halko _et al_ "Finding structure with randomness: Probabilistic algorithms for constructing approximate matrix decompositions." SIAM review 53.2 (2011): 217-288.
>
>
> > Explain how the error bars (like those in Figure 2) are generated. [...] Explain why some methods have error bars while some do not (in Figure D, and other figures, PCA does not have error bars while TLDR does).
>
> The proposed method has some inherent stochasticity, e.g. from SGD or from the neighbor pair sampling step (Step 2 in Algorithm 1). To make sure we properly measure variance, we run each variant of TLDR shown in Figures 2 and 6 _for five times_ and average the output results; the error bars report the standard deviation across those 5 runs. The reason we only report error bars for TLDR is because it has some noticeable variance; all other methods either correspond to deterministic algorithms or had negligible variance and therefore the error bars are not visible. _We thank the reviewer and we will clarify this in the updated version_.

---

> > ### Comment · Reviewer_j6W2 · 2022-06-11
> > **Re: Response**
> >
> > Thank you for engaging with the reviews and making the changes to the manuscript!

---

### Review · Reviewer_hxnp · 2022-05-11

**Summary Of Contributions:**

This paper proposes a method for dimensionality reduction called TLDR based on the Barlow Twins technique for self-supervised learning (Zbontar et al., 2021). Contemporary self-supervised learning approaches for vision typically aim to learn representations that are consistent when the input image is transformed by e.g. cropping, flips, etc. TLDR uses a similar idea but replaces image transformations with nearest neighbors as computed by Euclidean distance in the input space. Thus representations should be consistent not when image transformations are applied (as in self-supervised learning) but rather when an input is replaced by one of its neighbors.
Experiments demonstrate that TLDR produces useful low-dimensional representations for image and document retrieval.

**Broader Impact Concerns:**

I do not believe there to be significant ethical implications of this work. The primary value of this work is to speed up predictions for downstream tasks by preserving meaningful information in a low-dimensional representation. This method in general does not appear to enable predictions beyond what could have already been done with the original data given enough computational resources.

**Requested Changes:**

- [Not critical] Clearly demonstrate benefit of TLDR vs. baselines in terms of scalability. For the limiting resource (whether it be time, memory, etc.) how do baselines such as LLE, LTSA, etc. compare to TLDR as a function of dataset subsampling percentage and/or dimensionality?
- [Not critical] Algorithm 1 is too high level and would benefit by taking a step closer to pseudocode (for example, it is not entirely how to create positive pairs by sampling).
- [Not critical] Add a discussion about how the general idea of modifying self-supervised learning by replacing image transformations with neighbors on a graph could be more generally applicable. In particular, how important is the Barlow Twins method to this work? Could a similar dimensionality reduction technique be created by applying the general idea on top of SwAV, SimCLR, SimSiam, etc.? Also, what other applications involving a neighbor graph (beyond dimensionality reduction) could benefit from a similar approach?

**Strengths And Weaknesses:**

Strengths:
- The TLDR framework is intuitive and easy to understand.
- This paper forges a link between self-supervised learning and manifold learning. I could see this being a fruitful connection, by applying methods in one area to the other.
- The method and experiments are clearly presented.
- Limitations of the method and where it is expected to do well are honestly discussed.
- Experimental results of the method are fairly strong, particularly those that show an improvement of the low-dimensional TLDR representations against the learned representations of the original method.
- Breadth of experiments across different tasks and datasets.
- Ablations show which hyperparameters are important and which are not.

Weaknesses:

- Relative to other manifold learning methods, the lack of operations such as eigendecomposition and complicated optimization is pointed to as a benefit of TLDR. There is not much evidence to back up this claim, except for statements in §3.3 about the extent to which the dataset had to be subsampled for other manifold learning methods.
- The motivation for specifically linear encoders could be better. If a simple MLP were to perform significantly better than a linear encoder, I doubt the extra complexity of the MLP would be a significant barrier to adoption.

---

> ### Author Response · Authors · 2022-05-16
> **Response to Reviewer hxnp  (1/2)**
>
> We thank the reviewer for their positive and constructive review. Below we provide some clarifications and respond to the reviewer’s questions. We welcome any further discussion. We intend to upload an updated version of the manuscript soon.
>
> __Detailed Responses__
>
> > Relative to other manifold learning methods, the lack of operations such as eigendecomposition and complicated optimization is pointed to as a benefit of TLDR. There is not much evidence to back up this claim [...]  For the limiting resource (whether it be time, memory, etc.) how do baselines such as LLE, LTSA, etc. compare to TLDR as a function of dataset subsampling percentage and/or dimensionality?
>
> This is a good point. In the tables below (and in the draft figure we uploaded [at this url](https://imgur.com/a/LuqFUhm)) we provide quantitative evidence by measuring the training time and ROxford performance for TLDR and for a few manifold learning methods as a function of the dataset subsampling (we subsampled Google Landmarks v2-clean dataset, 1.5 million images in total) for output dimension $d=32$. All methods are run on the same servers. We used 16 CPUs and 300GB memory for all manifold learning methods, and one 32GB V100 GPU for TLDR. There is no publicly available easy-to-use GPU implementation for these methods, so we instead use the multi-core versions of scikit-learn.
>
> The tables show that methods like LLE and LTSA scale exponentially with respect to both the time and memory needed. When sampling 5% of the data, i.e. 75k images, TLDR (and LPP) require 5 minutes to train while LLE and LTSA require over 4 hours. The LLE and LTSA runs we started 4 days ago are still running when subsampling 8% or 10% of the data, while subsampling 15% and above led to out-of-memory (OOM) related crashes despite 300GB of memory available. The LPP method is scaling better, but a) is 5x slower than TLDR when subsampling 10% of the dataset and b) leads to OOM when subsampling 15%. _We will add these timings in the appendix of the updated version_.
>
> Please note also that Figure D in the Appendix further shows how **TLDR performance highly benefits from larger training sets, a property not shared with PCA**. As we see from this figure, PCA outperforms TLDR for a highly reduced number of images, however, it does not benefit from adding more data. In contrast, TLDR does benefit from adding more image. The plot even suggest that an even larger training set could potentially increase the superiority of TLDR vs PCA.
>
> Obviously, reported times highly depend on their implementation. Here, we used the popular scikit-learn multi-core implementations for LLE and LTSA and the best implementation of LPP we could find (https://github.com/jakevdp/lpproj). For TLDR we used our own pytorch codebase that we **will make publicly available with an easy-to-use interface** mimicking the scikit-learn API.
>
> __ROxford-Mean__ (mAP) | 1% | 2% | 3% | 5% | 8% | 10% | 15% | 20% | 30%
> -- | -- | -- | -- | -- | -- | -- | -- | -- | --
> PCA-w (16 CPU) | 0.362 | 0.368 | 0.377 | 0.377 | 0.365 | 0.362 | 0.359 | 0.363 | 0.370
> LLE (16 CPU) | 0.041 | 0.043 | 0.063 | 0.083 | - | - | X | X | X
> LTSA (16 CPU) | 0.077 | 0.061 | 0.071 | 0.075 | -  | - | X | X | X
> LPP (16 CPU) | 0.349 | 0.367 | 0.343 | 0.387 | 0.367 | 0.402 | X | X | X
> TLDR (1 GPU) | 0.370 | 0.375 | 0.384 | 0.404 | 0.418 | 0.414 | 0.427 | 0.426 | 0.426
>
>
> __Time (sec)__ | 1% | 2% | 3% | 5% | 8% | 10% | 15% | 20% | 30%
> -- | -- | -- | -- | -- | -- | -- | -- | -- | --
> PCA-w (16 CPU) | 4.429 | 5.973 | 9.063 | 15.098 | 20.123 | 23.814 | 38.158 | 52.195 | 86.909
> LLE (16 CPU) | 190.105 | 1210.304 | 3878.904 | 15963.771 | running 4 days | running 4 days | MEM LIMIT | MEM LIMIT | MEM LIMIT
> LTSA (16 CPU) | 678.068 | 1066.653 | 5095.748 | 15536.903 | running 4 days | running 4 days | MEM LIMIT | MEM LIMIT | MEM LIMIT
> LPP (16 CPU) | 12.142 | 50.183 | 95.863 | 297.192 | 1126.191 | 2250.844 | MEM LIMIT | MEM LIMIT | MEM LIMIT
> TLDR (1 GPU) | 213.736 | 233.985 | 259.251 | 311.693 | 375.207 | 420.121 | 541.407 | 641.811 | 874.099

---

> > ### Author Response · Authors · 2022-05-16
> > **Response to Reviewer hxnp (2/2)**
> >
> >
> >
> > > The motivation for specifically linear encoders could be better. If a simple MLP were to perform significantly better than a linear encoder, I doubt the extra complexity of the MLP would be a significant barrier to adoption.
> >
> > We focused on linear encoder a) to be directly comparable to PCA when it comes to encoding time, and b) because this is the setting used in practice for large-scale retrieval systems, i.e. the main application we tested. We also evaluated MLPs for some of the tasks and datasets (e.g. in Figure 2), and observed limited gains.
> > One reason could be that the features we use are highly optimized for the task. We agree that in the future, other applications of TLDR may find the use of MLPs more suited to their task, which is why our code allows for this option.
> >
> > > Algorithm 1 is too high level and would benefit by taking a step closer to pseudocode (for example, it is not entirely how to create positive pairs by sampling).
> >
> > This is a great suggestion. __We will add pytorch-style pseudo-code of TLDR in the appendix of the updated version__, also provided here [as a screenshot](https://imgur.com/a/LuqFUhm)
> >
> > > Add a discussion about how the general idea of modifying self-supervised learning by replacing image transformations with neighbors on a graph could be more generally applicable. [...] what other applications involving a neighbor graph (beyond dimensionality reduction) could benefit from a similar approach?
> >
> > As we discuss in Section 4 (paragraph “TLDR out of its comfort zone”), we do not consider neighbors to be a replacement for image transformations for learning large visual  representation models from scratch. We however believe that this signal can be useful as another way of performing model distillation, by e.g. sampling neighbors from a larger model or a model trained on more data. __We thank the reviewer and will mention this in the updated manuscript._
> >
> > > In particular, how important is the Barlow Twins method to this work? Could a similar dimensionality reduction technique be created by applying the general idea on top of SwAV, SimCLR, SimSiam, etc.?
> >
> > This is a very interesting point. It is true that we could devise a variant of TLDR that uses other “symmetrical” losses, for example SimCLR, SimSiam or VICReg. Yet, variants inspired by EMA like BYOL or MoCo might not be trivial. We however believe that the Barlow Twins (BT) loss is an exceptional fit given that the second term not only gives BT an elegant way of avoiding collapse (eg over SimSiam), but de-correlating the output space is a process that highly suits the dimensionality reduction task. Moreover, from our experiments and ablations we saw that the BT loss was highly robust to most hyperparameters (see ablations in the main paper and the Appendix). We will add this discussion in Section 4.

---

> > > ### Comment · Reviewer_hxnp · 2022-06-10
> > > **Re: Response**
> > >
> > > Thank you for your response. The new timing benchmarks help to more clearly demonstrate the benefits of the proposed method. The extended discussion section also better situates the method with respect to previous work.

---

### Author Response · Authors · 2022-05-24
**Updated version**

We thank all reviewers once again for their constructive reviews. We just __uploaded an updated version__ of our manuscript that includes responses to all comments and questions by the three reviewers. We are grateful for their thoughtful reviews that we believe made the manuscript clearer and stronger. Note that because of the additions below, the current manuscript is now more than 12 pages long (approx. 14 pages).

New or updated text is color-coded in blue for review convenience.

Specifically, the updated version features the following changes:
* Introduction (Sec 1)
  * Rewrite and rephrase parts to clarify the goal of the method proposed in our manuscript, emphasizing that it does not tackle (self-supervised) representation learning, but dimensionality reduction [FuBN]
* Experiments (Sec 3)
  * Added section “3.1 Summary of tasks and the evaluation protocol” with details on the tasks we solve and the evaluation process [FuBN]
  * Add implementation details on PCA and discuss its scalability
  * Added a paragraph on variance and the error bars of TLDR in the figures [j6W2]
* We split the “Discussion and related work (Sec 4)” into two sections for clarity:
  * 4. Related work
    * Added a paragraph on relation to Deep CCA paper[FuBN]
    * Added a paragraph on papers that use temporal neighborhoods [FuBN]
  * 5. Discussion
    * Added a  paragraph discussing other SSL losses beyond BT [FuBN, hxnp]
    * Added a paragraph discussing possible use cases of TLDR beyond dimensionality reduction (eg distillation) [hxnp] and for distance
metrics beyond Euclidean [FuBN]
    * Added a paragraph on training time and computational complexity [FuBN, j6W2, hxnp]
    * Added a paragraph clarifying “scalability” [FuBN, j6W2, hxnp]
* Appendix:
  * Add Appendix B.6 with discussion and a figure with timing comparisons  [hxnp]
  * Add pytorch-style pseudo-code in Appendix F. [hxnp]

---

### Decision · Action_Editors · 2022-06-13

**Recommendation:** Accept as is

**Comment:**

Thanks for your submission to TMLR.

The reviewers are all in agreement that the paper is ready for publication and the concerns have been sufficiently addressed by the authors.  I'm happy to recommend acceptance as-is for this paper.   I appreciate both the thoroughness of the responses/edits, as well as the efficiency in responding.